# Arcuate nucleus and lateral hypothalamic CART neurons in the mouse brain exert opposing effects on energy expenditure

Aitak Farzi[1,2†‡], Jackie Lau[1†], Chi Kin Ip[1†], Yue Qi[1], Yan-Chuan Shi[1], Lei Zhang[1], Ramon Tasan[3], Günther Sperk[3], Herbert Herzog[1*]

[1]Neuroscience Division, Garvan Institute of Medical Research, Sydney, Australia; [2]Otto Loewi Research Center, Pharmacology Section, Medical University of Graz, Graz, Austria; [3]Department of Pharmacology, Medical University Innsbruck, Innsbruck, Austria

**\*For correspondence:**
h.herzog@garvan.org.au

[†]These authors contributed equally to this work

**Present address:** [‡]Otto Loewi Research Center, Pharmacology Section, Medical University of Graz, Graz, Austria

**Competing interests:** The authors declare that no competing interests exist.

**Abstract** Cocaine- and amphetamine-regulated transcript (CART) is widely expressed in the hypothalamus and an important regulator of energy homeostasis; however, the specific contributions of different CART neuronal populations to this process are not known. Here, we show that depolarization of mouse arcuate nucleus (Arc) CART neurons via DREADD technology decreases energy expenditure and physical activity, while it exerts the opposite effects in CART neurons in the lateral hypothalamus (LHA). Importantly, when stimulating these neuronal populations in the absence of CART, the effects were attenuated. In contrast, while activation of CART neurons in the LHA stimulated feeding in the presence of CART, endogenous CART inhibited food intake in response to Arc CART neuron activation. Taken together, these results demonstrate anorexigenic but anabolic effects of CART upon Arc neuron activation, and orexigenic but catabolic effects upon LHA-neuron activation, highlighting the complex and nuclei-specific functions of CART in controlling feeding and energy homeostasis.
DOI: https://doi.org/10.7554/eLife.36494.001

## Introduction

The neuropeptide cocaine- and amphetamine-regulated transcript (CART) is encoded by *Cart pre-propeptide (Cartpt)* and involved in the regulation of a diverse range of physiological functions including food intake and energy homeostasis, thermoregulation, reward as well as stress processing (*Lau and Herzog, 2014*; *Subhedar et al., 2014*). While the lack of knowledge on the potential CART receptor(s) hampers insights into the detailed mechanistic processes underlying CART actions, the phenotypes of *Cartpt*-knockout (KO) mice and the effects of intracerebroventricular (i.c.v.) injection of CART peptides suggest anorexigenic and catabolic actions of CART (*Lau and Herzog, 2014*; *Kristensen et al., 1998*; *Lau et al., 2016*). In contrast, the administration of CART into specific hypothalamic nuclei demonstrated orexigenic effects (*Abbott et al., 2001*), indicating location-dependent functions of CART.

In the hypothalamus, both the arcuate nucleus (Arc) and the lateral hypothalamic area (LHA) show high expression of CART and play a central role in a variety of homeostatic functions, including the regulation of energy homeostasis (*Lau and Herzog, 2014*). The Arc represents the primary site of the coordination of peripheral signals to integrate nutritional status of the body with other central pathways controlling energy homeostasis, including the ones in the LHA (*Barsh and Schwartz, 2002*). Classically, two populations of neurons located in the Arc are regarded as the main regulators of energy homeostasis, the orexigenic neuropeptide Y (NPY)/agouti-related peptide (AgRP) neurons and the anorexigenic proopiomelanocortin (POMC) neurons, which to some extent coexpress CART

(*Elias et al., 1998a*,*1998b*; *Campbell et al., 2017*). The LHA, in turn, receives projections from the Arc (*Elias et al., 1999*) and is generally regarded as a critical regulator of feeding, motivated behavior and arousal, given that electrical stimulation of the LHA promotes feeding and reinforcement processes, whereas lesions of the LHA decrease ingestive behavior (*Stuber and Wise, 2016*). In addition, in situ hybridization studies show high coexpression of *Cartpt* with the orexigenic melanin-concentrating hormone (MCH, encoded by *Pmch*) and glutamate decarboxylase (GAD67, encoded by *Gad1*) in the LHA (*Vrang, 2006*), supporting the anabolic actions of LHA CART (*Lau et al., 2018*). Importantly, recent findings from our laboratory show that CART neuron-specific reintroduction of CART in the Arc or LHA in otherwise CART-deficient mice exerts differential effects on energy expenditure, while food intake was not affected (*Lau et al., 2018*), highlighting the importance and differential responses to regional actions of CART.

In this study, we aimed to investigate the contribution of CART neuron activation to the regulation of energy expenditure focusing on the Arc and LHA, respectively. Activation of CART neurons was achieved by the use of the designer receptors exclusively activated by designer drugs (DREADD) technology (*Alexander et al., 2009*) delivered by a *Cre-recombinase*-dependent adeno-associated viral (AAV) vector to specific hypothalamic nuclei of *Cartpt-cre* knock-in mice. Comparison of the metabolic responses of $Cartpt^{cre/cre}$ mice (which do not express CART) and $Cartpt^{cre/+}$ mice provides specific insights into the direct contribution of CART to the effects of CART neuron activation at these two hypothalamic nuclei.

## Results

### Confirmation of targeted injections of AAV-hM3Dq-mCherry into hypothalamic CART neurons

To verify targeted expression and functionality of the AAV-hM3Dq-mCherry construct, the location of the expression of the DREADD-containing mCherry reporter and the neuronal activation upon stimulation with clozapine-*N*-oxide (CNO) defined by c-Fos expression were determined. Fluorescence microscopy confirmed that in $Cartpt^{cre/+}$ mice stereotaxically injected with AAV-hM3Dq-mCherry into the Arc (*Figure 1A*) or LHA (*Figure 1B*), the expression of mCherry is location-specific. This evaluation procedure was also used as the basis to exclude mice from further analysis if they did not fit the exact injection criteria and only data from mice identified for positive hits were used for downstream analysis.

To examine the functionality of the DREADD vector, immunostaining for c-Fos expression was performed. Brain slices from Arc→hM3Dq or LHA→hM3Dq $Cartpt^{cre/+}$ mice subjected to a single i.p. injection of CNO 60 min prior to sacrifice were used (*Han et al., 2017*). *Figure 1C and D* show extensive c-Fos immunoreactivity in the respective targeted areas in response to CNO injection, confirming the functionality of the approach, while no c-Fos immunoreactivity could be observed 60 min after saline injection (*Figure 1C,D*).

### Activation of Arc→hM3Dq CART neurons suppresses energy expenditure and physical activity in the presence of endogenous CART

In order to investigate the contribution of Arc CART to energy expenditure regulation, we unilaterally injected 10-week-old CART-expressing $Cartpt^{cre/+}$ as well as CART-deficient $Cartpt^{cre/cre}$ mice with the stimulatory hM3Dq vector into the Arc (Arc→hM3Dq). Parameters of energy expenditure, physical activity and respiratory exchange ratio (RER) under free-feeding conditions were measured using open-circuit indirect calorimetry. The depolarization of the Arc CART neurons was achieved by i.p. injection of CNO and the same mice were used as their own control injected with saline 24 hr later.

Interestingly, while no change in energy expenditure was seen following activation of CART neurons in Arc→hM3Dq $Cartpt^{cre/cre}$ mice (*Figure 2A*), energy expenditure was significantly decreased 3–6 hr (08:00–11:00 pm) after CNO injection in Arc→hM3Dq $Cartpt^{cre/+}$ mice (*Figure 2B*) compared to saline-injected controls. The decrease in energy expenditure was driven by the decrease of both, $VO_2$ and $VCO_2$ (*Figure 2B*). To account for potential unspecific vector expression and effects of CNO treatment on its own, wild type mice that do not express *Cre-recombinase* were injected with the stimulatory hM3Dq vector as well as $Cartpt^{cre/+}$ and $Cartpt^{cre/cre}$ mice injected with empty vector, all receiving i.p. CNO, were used as an additional control. Importantly, no effects of CNO on its

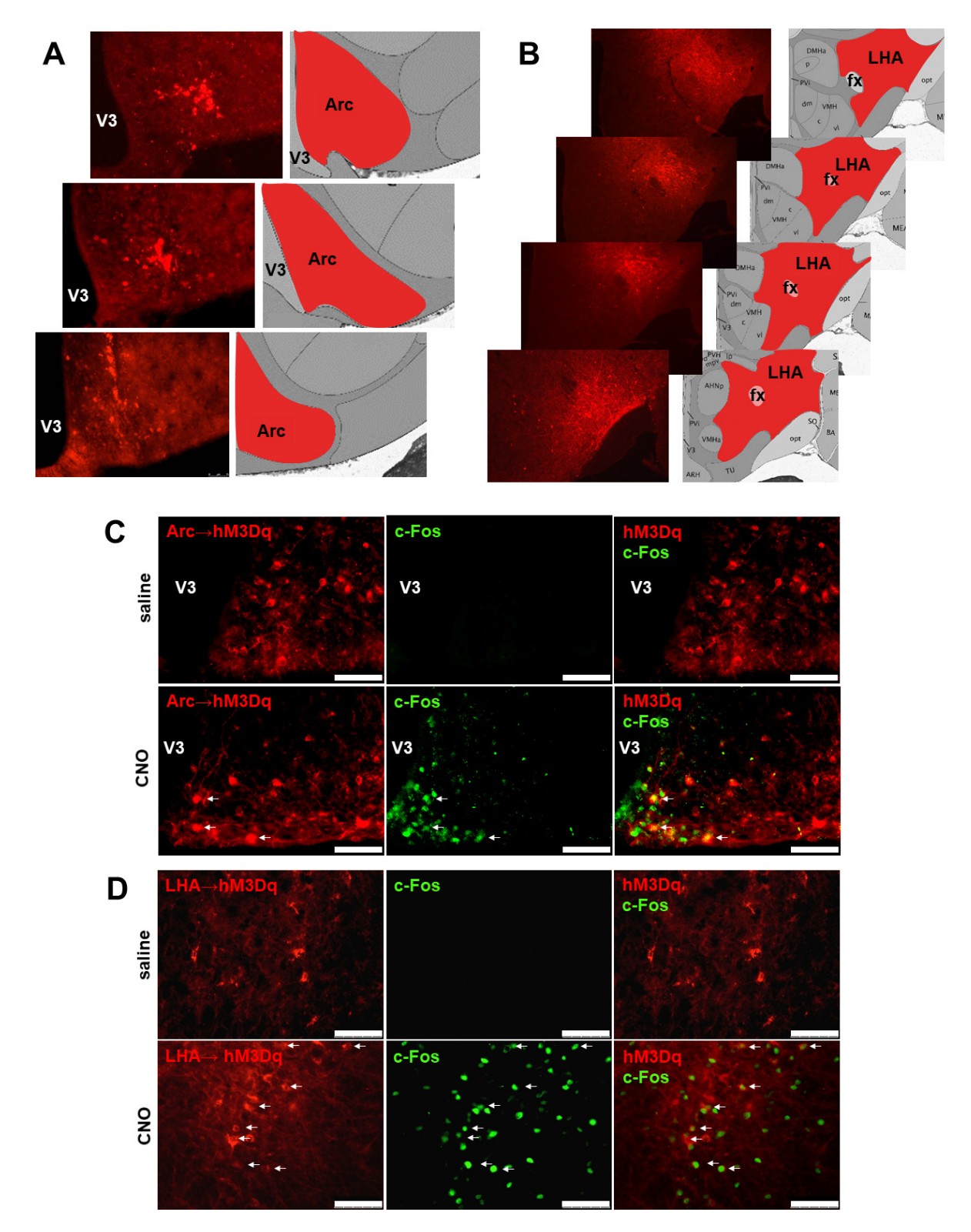

**Figure 1.** Validation of the expression sites of C*re*-dependent *AAV-hM3Dq-mCherry* for CNO-mediated neuronal activation. Fluorescence micrographs of coronal brain sections from respective Arc→hM3Dq (**A**) and LHA→hM3Dq (**B**) *Cartpt^cre/+* mice, showing the mCherry reporter being specifically expressed in the respective areas. Selective CNO-mediated neuronal activation marked by c-Fos expression in hM3Dq-mCherry-positive CART neurons at the Arc (**C**) and the LHA (**D**) (white arrows). Scale bar = 75 µm. Arc, arcuate nucleus; fx, fornix; LHA, lateral hypothalamic area; V3, third ventricle.

*Figure 1 continued on next page*

*Figure 1 continued*

DOI: https://doi.org/10.7554/eLife.36494.002

own were evident in these controls in any of the parameters under study (*Figure 2—figure supplement 1*). This indicates that CART signaling in this neuronal population is essential and directly involved in the generation of these effects.

A similar pattern was observed for physical activity where locomotion in Arc→hM3Dq *Cartpt^{cre/cre}* mice was unaltered in response to CNO activation (*Figure 2C*), while CART-positive Arc→hM3Dq *Cartpt^{cre/+}* cohorts displayed a decrease in locomotion upon CNO treatment compared to saline controls (*Figure 2D*). In line with the downregulated energy expenditure, the decrease in locomotion mainly spanned the period of 2–6 hr (07:00 – 11:00 pm) post-CNO injection. Investigation of fuel source preference indexed by the respiratory exchange ratio (RER) showed no noticeable difference between saline and CNO treatments throughout the light and dark photoperiods, in either genotype (*Figure 2E,F*).

To further investigate the dependence on endogenous CART for the decrease in energy expenditure, we measured body temperature parameters in response to CNO treatment in these mice by high-sensitivity infrared imaging. Importantly, while no change of temperature was observed in Arc→hM3Dq *Cartpt^{cre/cre}* mice (*Figure 3A*), a significant decrease in the skin temperature above the area of the brown adipose tissue (BAT) and lumbar back region was detected in the Arc→hM3Dq *Cartpt^{cre/+}* mice after CNO treatment (*Figure 3B*). This is consistent with the observed decrease in energy expenditure in the presence of CART in these mice (*Figure 2B*). In order to assess the contribution of BAT to overall body temperature, we calculated the difference between BAT and lumbar temperatures. CNO-induced Arc→hM3Dq CART neuron activation did not lead to a change in the differences between BAT and lumbar temperatures compared to saline controls (*Figure 3A,B* right panels), indicating that both BAT and lumbar temperature decreased to the same extent upon Arc→hM3Dq CART neuronal activation. These results suggest that the observed decrease in skin temperature is not due to a specific change in BAT thermogenic activity. Furthermore, the observation that the decrease in temperature is paralleled by a decrease in physical activity suggests that decreased activity-related thermogenesis may at least in part contribute to the observed temperature changes.

## Activation of Arc →hM3Dq CART neurons increases 24 hr food intake and body weight gain in the absence of endogenous CART

Given that endogenous CART affected energy expenditure and body temperature in response to Arc→hM3Dq CART neuron activation, effects of CNO-conferred activation of Arc→hM3Dq CART neurons on food consumption and body weight (BW) were also investigated in *Cartpt^{cre/+}* and *Cartpt^{cre/cre}* mice.

Spontaneous food intake in absolute amounts measured over 24 hr showed a trend of increase in CNO-treated Arc→hM3Dq *Cartpt^{cre/cre}* mice where CART is absent when compared to the same mice injected with saline (*Figure 4A*); this increase reached statistical significance when expressed as a percentage of BW (*Figure 4B*). In contrast, food intake was not affected in the *Cartpt^{cre/+}* mice both in absolute amounts (*Figure 4C*) as well as a percentage of BW (*Figure 4D*). In line with the effects seen on food intake, a significant weight gain (relative to pre-treatment BW) was observed in the Arc→hM3Dq *Cartpt^{cre/cre}* mice following CNO treatment (*Figure 4E*), while CNO-mediated stimulation of Arc→hM3Dq CART neurons did not induce a change in BW in the *Cartpt^{cre/+}* mice (*Figure 4F*). The lack of increase in food intake and BW gain in response to Arc→hM3Dq CART neuron activation in the presence of CART supports an inhibitory role of endogenous CART on food intake in response to activation of this neuronal population.

## Activation of LHA →hM3Dq CART neurons promotes energy expenditure and physical activity in the presence of endogenous CART

The LHA constitutes another pivotal component in the hypothalamic circuit for the regulation of energy balance, which receives axonal projections from the Arc and has been implicated with both anorectic and orexigenic potentials (*Stuber and Wise, 2016*). As we have recently discovered an

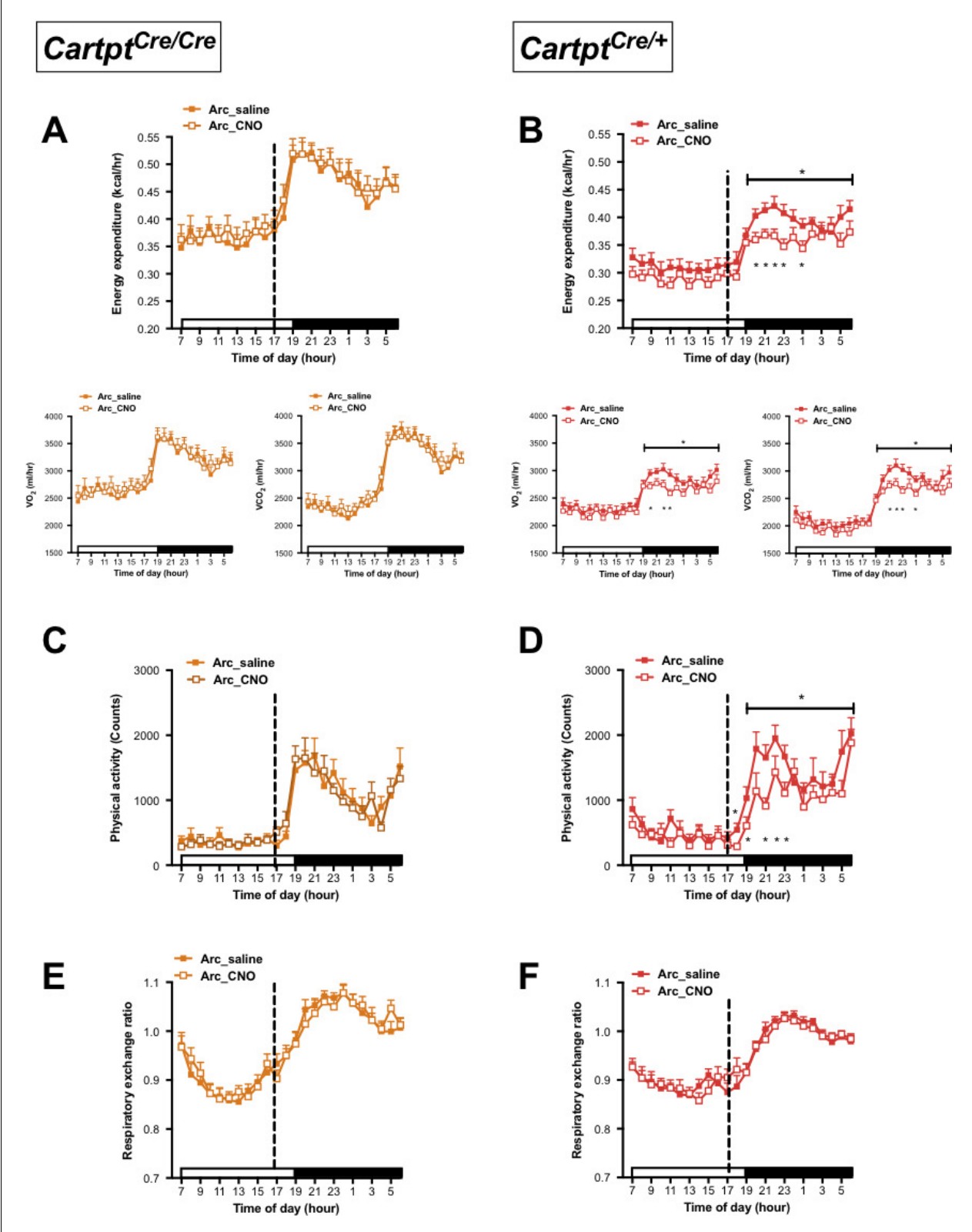

**Figure 2.** CNO-mediated activation of hM3Dq-containing Arc CART neurons leads to a reduction in energy expenditure and physical activity in *Cartpt*$^{cre/+}$ mice. Indirect calorimetric assessments following saline or CNO injection for the 24 hr time course of energy expenditure, VO$_2$ and VCO$_2$ (A, B), physical activity (C, D), and respiratory exchange ratio (E, F) in Arc→hM3Dq *Cartpt*$^{cre/cre}$ and Arc→hM3Dq *Cartpt*$^{cre/+}$ mice, respectively. Open and
*Figure 2 continued on next page*

*Figure 2 continued*

filled horizontal bars indicate the light and dark photoperiods, respectively. Dotted line indicates the time of i.p. injection of saline or CNO. Data are means ±SEM. n = 10–12; *p≤0.05 for saline versus CNO treatments.

DOI: https://doi.org/10.7554/eLife.36494.003

The following figure supplement is available for figure 2:

**Figure supplement 1.** CNO does not exert any effects on physical activity, energy expenditure and respiratory exchange ratio in the absence of hM3Dq-expression in *Cartpt^{cre/cre}*, *Cartpt^{cre/+}* and wild type mice.

DOI: https://doi.org/10.7554/eLife.36494.004

important role for LHA CART in regulating energy expenditure and body temperature (*Lau et al., 2018*), we were also interested in investigating the impact of endogenous CART upon LHA CART neuron activation employing the same technology as used for the investigation in the Arc.

While no effect on energy expenditure was observed in the CART-deficient LHA→hM3Dq *Cartpt^{cre/cre}* mice upon CNO stimulation (*Figure 5A*), LHA→hM3Dq *Cartpt^{cre/+}* mice, which express CART, demonstrated a significant CNO-induced increase in energy expenditure (both $VO_2$ and $VCO_2$) (*Figure 5B*), contrasting to the reduction observed in Arc→hM3Dq *Cartpt^{cre/+}* mice (*Figure 2B*). The elevation in energy expenditure in the presence of endogenous CART was especially evident during the dark phase between the hours of 07:00 pm – 04:00 am (*Figure 5B*), highlighting the specific involvement of LHA CART neurons in affecting energy expenditure in the period when the mice are more active.

Similar to the pattern seen for energy expenditure, activation of LHA CART neurons in LHA→hM3Dq *Cartpt^{cre/cre}* mice, which lack CART, had no significant effect on physical activity (*Figure 5C*). However, stimulation of LHA→hM3Dq *Cartpt^{cre/+}* mice with CNO, where endogenous CART is present, led to a significant increase in physical activity compared to saline treatment most prominently so within the time window when the increase in energy expenditure occurred, that is the early part of the dark phase (*Figure 5D*). Resembling the observations from the Arc→hM3Dq groups (*Figure 2E,F*), RER in the LHA→hM3Dq mice was unaltered between saline and CNO treatment regardless of genotype (*Figure 5E,F*).

As these results indicate a role for CART in heat dissipation in response to activation of this subset of CART neurons in the LHA, we were also interested in body temperature. Interestingly, however, no difference in BAT or lumbar temperature was detected in LHA→hM3Dq *Cartpt^{cre/cre}*

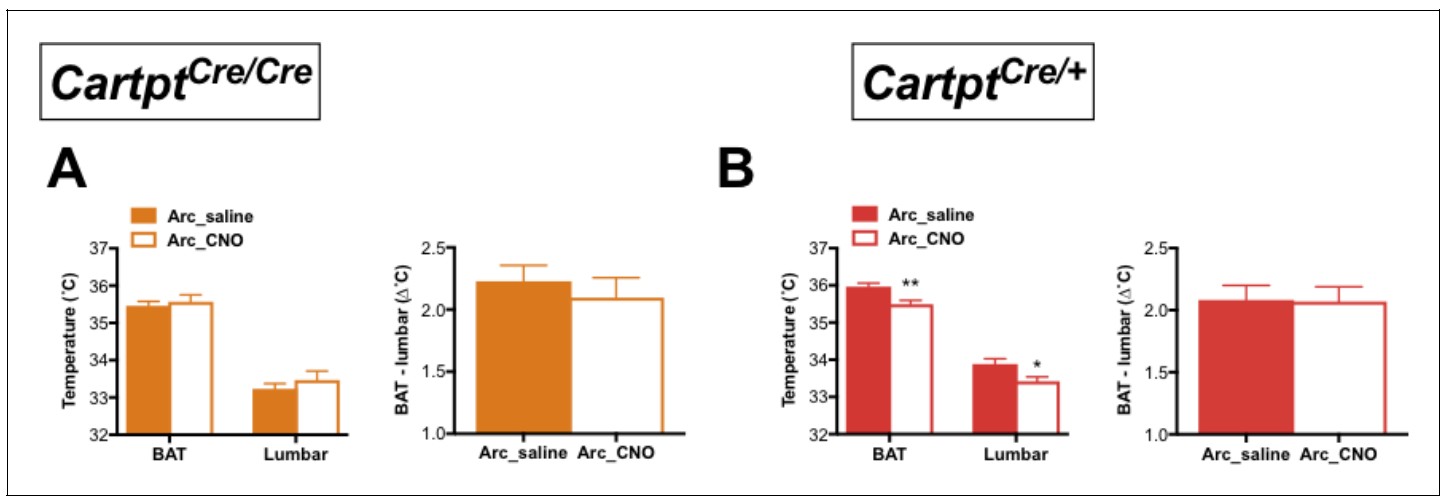

**Figure 3.** CNO-mediated activation of hM3Dq-containing Arc CART neurons leads to a decrease in body skin temperature in *Cartpt^{cre/+}* mice. Temperatures of the interscapular brown adipose tissue (BAT) and the lumbar back region, as well as the temperature differences between the interscapular and lumbar areas 2 hr following saline or CNO injection, measured by high-sensitivity infrared imaging in Arc→hM3Dq *Cartpt^{cre/cre}* (A) and Arc→hM3Dq *Cartpt^{cre/+}* mice (B). Data are means ±SEM. n = 10–12; *p≤0.05, **p≤0.01 for saline versus CNO treatments.

DOI: https://doi.org/10.7554/eLife.36494.005

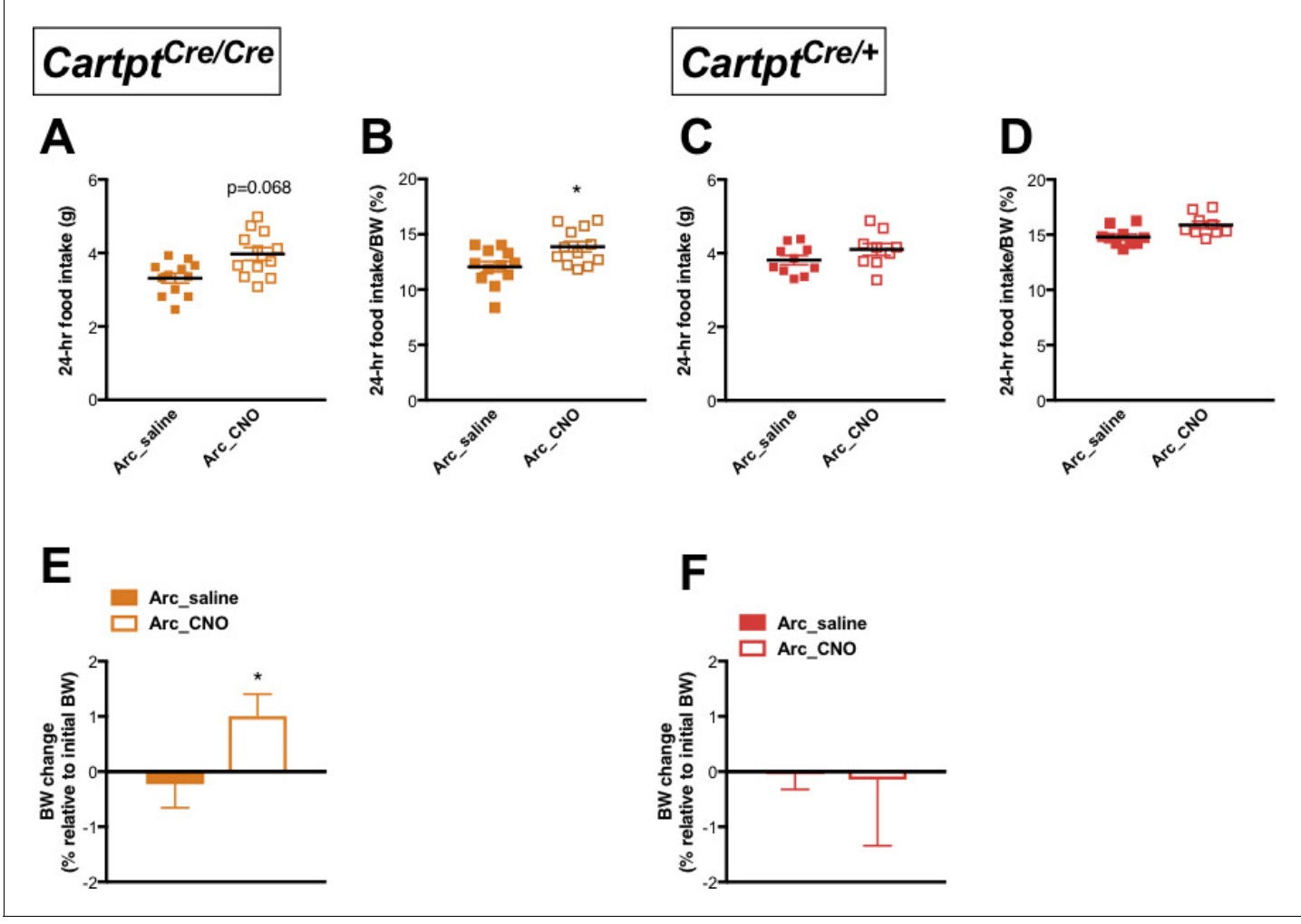

**Figure 4.** CNO-induced activation of hM3Dq-positive CART neurons in the Arc increases 24 hr food intake and results in weight gain in homozygous *Cartpt^{cre/cre}* mice. Food intake during fed state 24 hr following saline or CNO injection in Arc→hM3Dq mice, expressed both as absolute amounts of food intake (A, C) and as a percentage of body weight (B, D), for *Cartpt^{cre/cre}* and *Cartpt^{cre/+}* mice, respectively. The corresponding body weight change at 24 hr in proportion to pre-treatment body weight (E, F) measured in *Cartpt^{cre/cre}* and *Cartpt^{cre/+}* Arc→hM3Dq mice receiving saline or CNO injection. Data are means ±SEM. n = 10–12; *p≤0.05 for saline versus CNO treatments.

DOI: https://doi.org/10.7554/eLife.36494.006

(*Figure 6A*) or LHA→hM3Dq *Cartpt^{cre/+}* mice (*Figure 6B*) in response to LHA→hM3Dq CART neuron activation by CNO.

## Activation of LHA→hM3Dq CART neurons increases 24 hr food intake and body weight gain in the presence of endogenous CART

As we observed a role for endogenous CART in promoting energy expenditure in response to LHA CART neuron activation, we next investigated whether this also influences the other side of the energy balance equation, food intake. In comparison with saline treatment, CNO-induced stimulation of LHA CART neurons in LHA→hM3Dq *Cartpt^{cre/cre}* mice, which do not express CART, showed a trend to an increase in 24 hr food intake, both in absolute amounts (*Figure 7A*) as well as when expressed as a percentage of BW (*Figure 7B*). In contrast, CNO-mediated activation of CART neurons in LHA→hM3Dq *Cartpt^{cre/+}* mice significantly increased absolute (*Figure 7C*) as well as percentage of BW food intake (*Figure 7D*). Consistent with the effects on food intake, LHA→hM3Dq *Cartpt^{cre/cre}* mice showed no change in BW gain when injected with CNO compared to saline (*Figure 7E*), while CNO-treated LHA→hM3Dq *Cartpt^{cre/+}* mice displayed a significant increase in

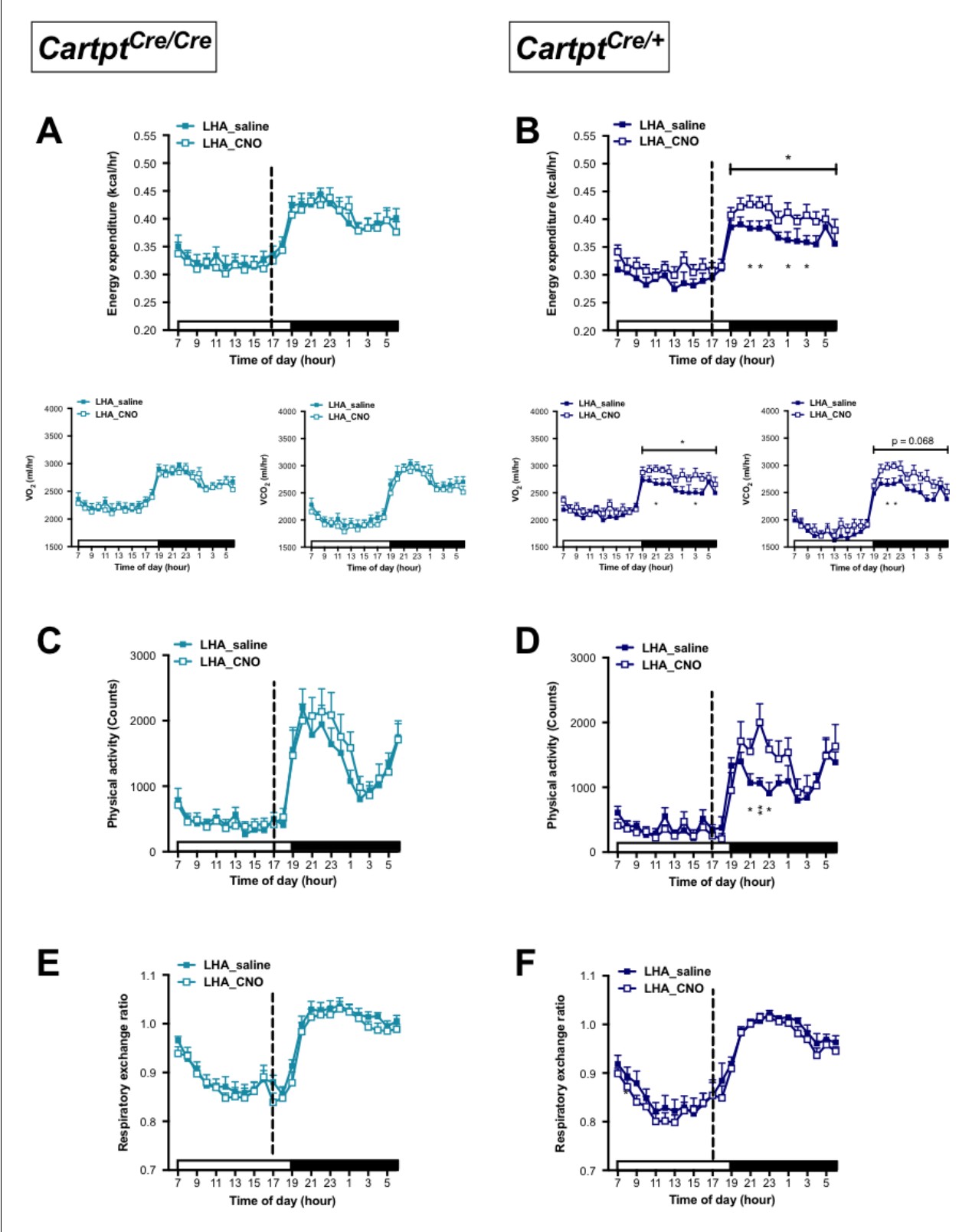

**Figure 5.** CNO-mediated activation of hM3Dq-containing LHA CART neurons increases energy expenditure and physical activity in *Cartpt^cre/+* mice. Indirect calorimetric assessments following saline or CNO injection for the 24 hr time course of energy expenditure, VO₂ and VCO₂ (A, B), physical activity (C, D), and respiratory exchange ratio (E, F) in LHA→hM3Dq *Cartpt^cre/cre* and LHA→hM3Dq *Cartpt^cre/+* mice, respectively. Open and filled

*Figure 5 continued on next page*

*Figure 5 continued*

horizontal bars indicate the light and dark photoperiods, respectively. Dotted line indicates the time of i.p. injection of saline or CNO. Data are means ±SEM. n = 10–12; *p≤0.05, **p≤0.01 for saline versus CNO treatments.

DOI: https://doi.org/10.7554/eLife.36494.007

BW gain (*Figure 7F*). Together, these results suggest the direct involvement of CART in LHA neurons for the control of feeding.

### Presence of endogenous CART expression promotes energy expenditure and physical activity in response to bilateral activation of LHA→hM3Dq CART neurons

While unilateral LHA→hM3Dq CART neuron activation already showed some interesting effects in alterations of energy expenditure parameters, we wondered whether bilateral DREADD injections would be even more informative. Furthermore, as the effects of CNO in promoting energy expenditure and locomotion were likely overlaid by the general increase of physical activity during the dark phase, to more clearly investigate this aspect the CNO injection in the bilateral LHA→hM3Dq cohorts was brought forward to the middle of the light phase (12:00 pm) when physical activity in mice is naturally lower (*Figure 8*). Results obtained with the doubled neuronal activation and altered circadian condition revealed a significant increase in energy expenditure in both genotypes, and importantly this increase was more pronounced in the CART-containing LHA→hM3Dq *Cartpt*cre/+ mice than the ones lacking CART (*Figure 8A,B*). Similar results were observed for changes in physical activity (*Figure 8C,D*), that is a more pronounced increase in activity upon CNO activation in the presence of endogenous CART (*Figure 8D*).

Interestingly, while no change in RER in response to CNO was observed in LHA→hM3Dq *Cartpt*cre/cre mice (*Figure 8E*), bilaterally injected LHA→hM3Dq *Cartpt*cre/+ mice where CART is present did display an increase in RER following CNO treatment (*Figure 8F*), a marker of carbohydrate preference as oxidative fuel.

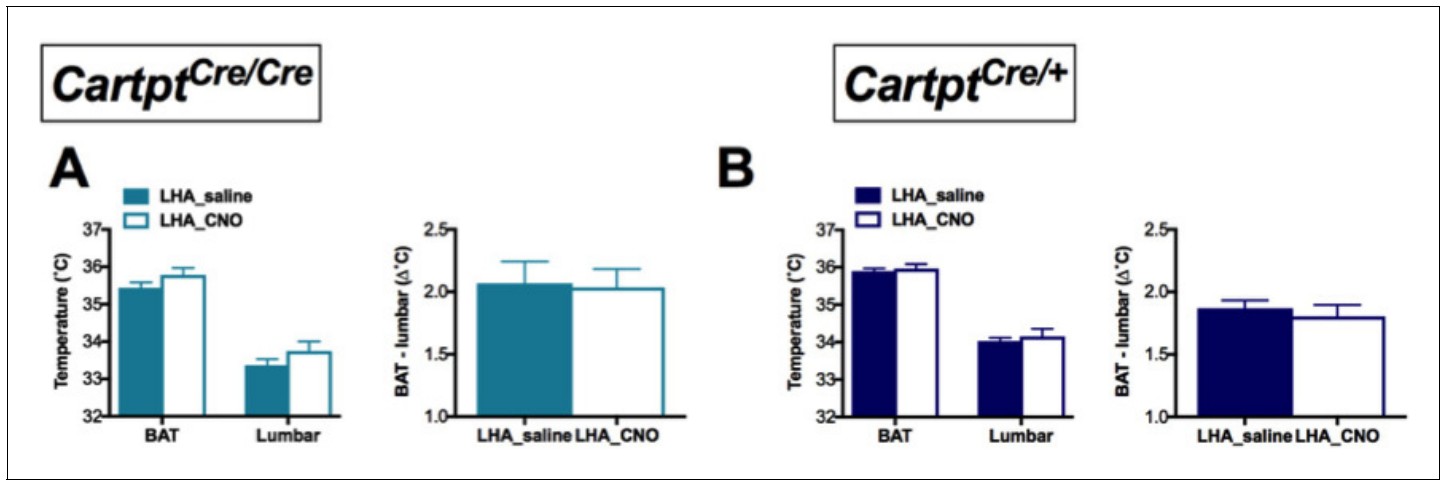

**Figure 6.** CNO-mediated unilateral activation of hM3Dq-containing LHA CART neurons does not affect body temperature of *Cartpt*cre/cre and *Cartpt*cre/+ mice. Temperatures of the BAT and the lumbar back region, as well as the temperature differences between the inter-scapular and lumbar areas of *Cartpt*cre/cre (A) and *Cartpt*cre/+ (B) mice measured by high-sensitivity infrared imaging 2 hr after i.p. injection of saline or CNO. Data are means ±SEM. n = 10–12.

DOI: https://doi.org/10.7554/eLife.36494.008

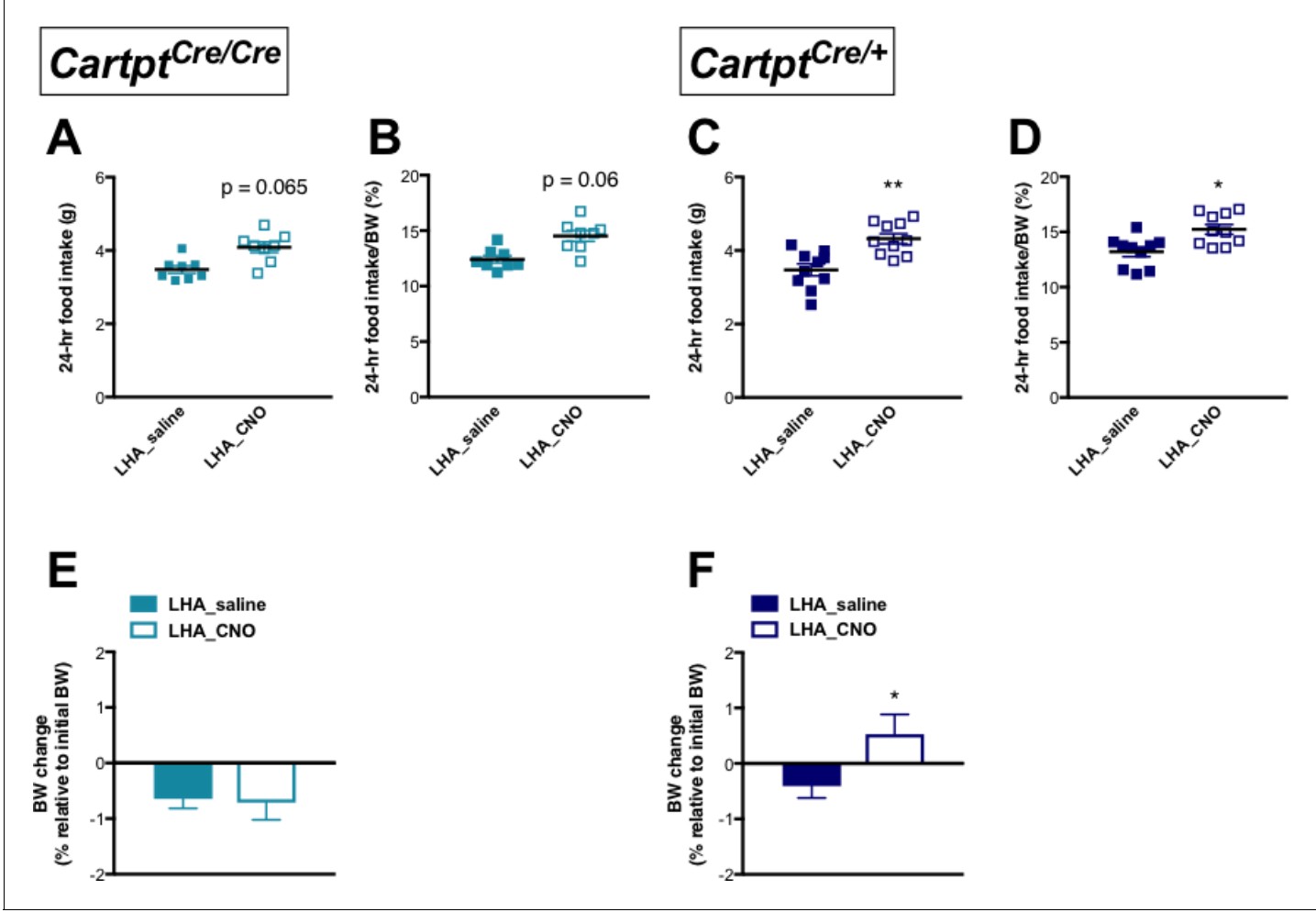

**Figure 7.** CNO-induced activation of hM3Dq-positive CART neurons in the LHA increases 24 hr food intake and body weight gain in heterozygous *Cartpt$^{cre/+}$* mice. Food intake during fed state 24 hr following saline or CNO injection in LHA$^{hM3Dq}$ mice, expressed both as absolute amounts (**A, C**) and as a percentage of body weight (**B, D**) for *Cartpt$^{cre/cre}$* and *Cartpt$^{cre/+}$* mice, respectively. The corresponding body weight change at 24 hr in proportion to pre-treatment body weight (**E, F**) measured in *Cartpt$^{cre/cre}$* and *Cartpt$^{cre/+}$* LHA→hM3Dq mice receiving saline or CNO injection. Data are means ±SEM. n = 10–12; *p≤0.05, **p≤0.01 for saline versus CNO treatments.
DOI: https://doi.org/10.7554/eLife.36494.009

## Presence of endogenous CART expression enhances BAT and body temperature in response to bilateral activation of LHA→hM3Dq CART neurons

In light of the stronger metabolic effects seen in the bilaterally compared to the unilaterally injected LHA→hM3Dq cohorts, we revisited the investigation of body temperature changes after LHA→hM3Dq CART neuron activation. Whereas no significant effects on body temperatures were seen upon unilateral activation of LHA→hM3Dq CART neurons by CNO (*Figure 6*), bilateral LHA→hM3Dq CART neuron activation induced a significant increase in BAT (*Figure 9A,B*) and lumbar (*Figure 9D,E*) skin temperatures in both LHA→hM3Dq *Cartpt$^{cre/cre}$* and LHA→hM3Dq *Cartpt$^{cre/+}$* mice, suggesting that the activation of a greater number of LHA CART neurons is required to exert body temperature effects. Interestingly, while the elevated BAT and lumbar temperatures started to decline in the LHA→hM3Dq *Cartpt$^{cre/cre}$* mice at 6 hr post-injection (*Figure 9A,D*), they remained elevated in the LHA→hM3Dq *Cartpt$^{cre/+}$* mice (*Figure 9B,E*), indicating a longer lasting effect in the presence of endogenous CART. Furthermore, comparison of BAT or lumbar temperature between *Cartpt$^{cre/+}$* and *Cartpt$^{cre/cre}$* mice in response to CNO injection (subtracting CNO

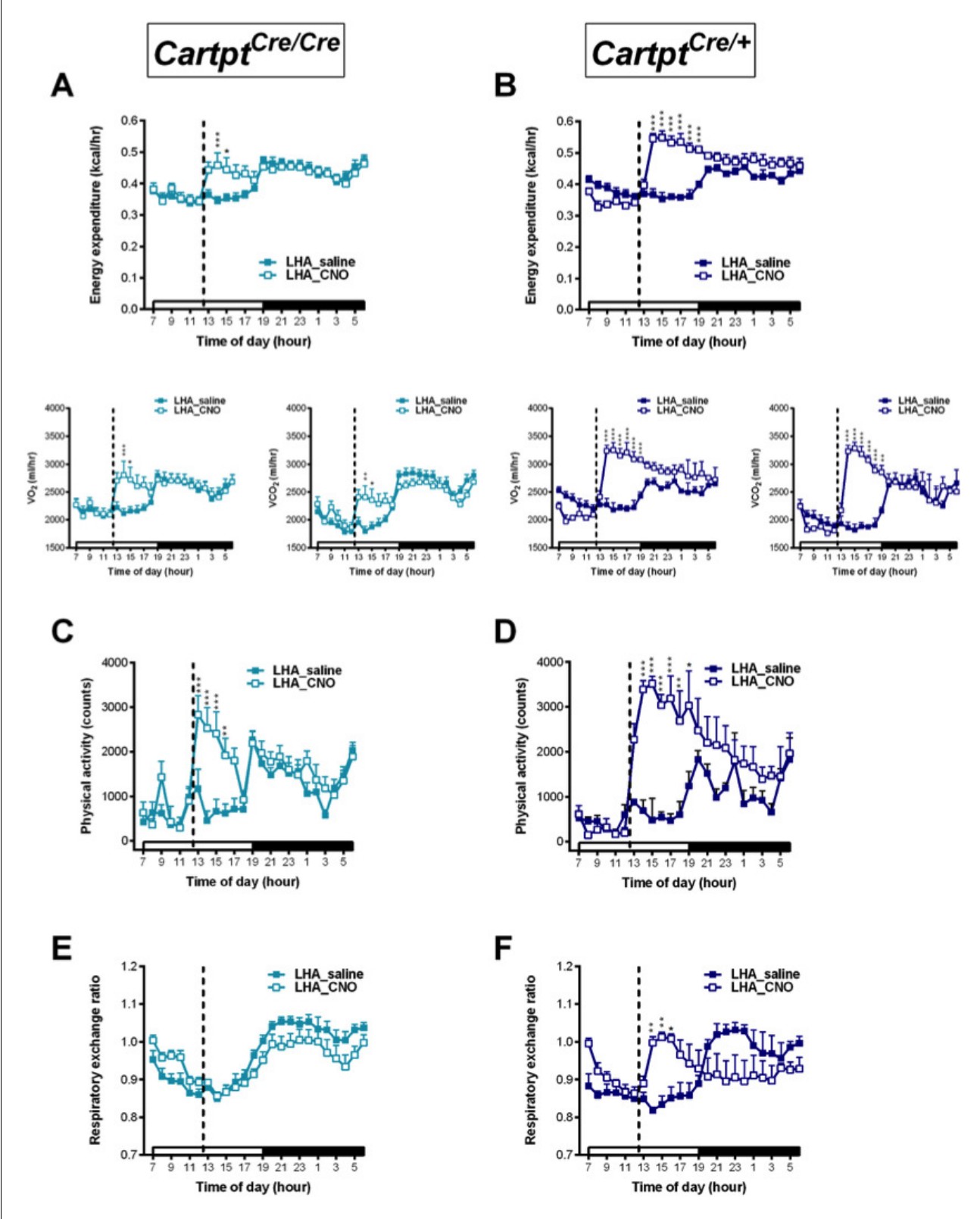

**Figure 8.** CNO-mediated bilateral activation of hM3Dq-containing LHA CART neurons leads to elevation in physical activity, energy expenditure and respiratory exchange ratio in *Cartpt^cre/+* mice. Indirect calorimetric assessments following saline or CNO injection for the 24 hr time course of energy expenditure, VO₂ and VCO₂ (**A, B**), physical activity represented as total count (**C, D**), and respiratory exchange ratio (**E, F**) in LHA→hM3Dq *Cartpt^cre/cre*

*Figure 8 continued on next page*

Figure 8 continued

and LHA→hM3Dq *Cartpt*<sup>cre/+</sup> mice, respectively. Open and filled horizontal bars indicate the light and dark photoperiods, respectively. Dotted lines indicate the time of i.p. injection of saline or CNO. Data are means ± SEM. n = 6–7; *p≤0.05, **p≤0.01, ***p≤0.001 for saline versus CNO treatments.
DOI: https://doi.org/10.7554/eLife.36494.010

values from saline at corresponding time-points for each mouse) demonstrates that the presence of CART promotes the increase in both BAT and lumbar temperatures (*Figure 9C,F*).

An increase in body temperatures could be due to increased thermogenesis or reduced heat loss, or both. Since the tail is a key regulator of heat dissipation in rodents, and a decrease in tail skin temperature is indicative of decreased heat loss, which could contribute to an increase in body temperature (*Fischer et al., 2016*) we also investigated tail temperature. Tail temperature was significantly decreased upon bilateral LHA→hM3Dq CART neuron activation by CNO in both LHA→hM3Dq *Cartpt*<sup>cre/cre</sup> and LHA→hM3Dq *Cartpt*<sup>cre/+</sup> mice (*Figure 9J,K*), indicating a contribution of decreased heat loss to the increased body temperatures by LHA CART neuron activation (*Figure 9A,B,D and E*). However, the decrease in tail temperature was not significantly different between *Cartpt*<sup>cre/+</sup> and *Cartpt*<sup>cre/cre</sup> mice (*Figure 9I*), suggesting that the stronger increase in BAT and lumbar temperatures in *Cartpt*<sup>cre/+</sup> mice is unlikely due to decreased heat loss through the tail, and other mechanism(s) are at play for the control of body temperature by endogenous CART in this neuronal population.

Since BAT is a thermogenic organ, we investigated the contribution of BAT functionality to the overall increase in body temperature by assessing the difference between BAT and lumbar temperatures, since during increased BAT thermogenesis the difference between BAT and lumber temperatures would be expected to increase due to direct contribution of heat arising from the BAT depot. Interestingly, there was a significant decrease in the differences between BAT and lumbar temperatures in response to CNO-induced LHA→hM3Dq CART neuron activation compared to saline controls (*Figure 9J,K*), indicating that lumbar temperature was increased to a greater extent than BAT temperature upon LHA→hM3Dq CART neuronal activation. These results suggest that BAT thermogenesis is not the major contributor to the observed increase in body temperature. Furthermore, the decrease in the difference between BAT and lumbar temperatures in *Cartpt*<sup>cre/+</sup> mice at 1 and 3 hr after CNO injection was greater than that in *Cartpt*<sup>cre/cre</sup> mice (*Figure 9L*), suggesting that lumbar back temperature is especially increased in *Cartpt*<sup>cre/+</sup> mice and that BAT thermogenesis is unlikely to contribute to the greater increase in body temperature induced by LHA→hM3Dq CART neuronal activation in the presence of endogenous CART. Control mice injected with CNO did not show any differences in thermogenic function confirming the specificity of the technique (*Figure 9—figure supplement 1*).

## Bilateral activation of LHA→hM3Dq CART neurons increases feeding behavior and body weight gain in the presence of endogenous CART

In order to examine whether bilateral LHA→hM3Dq application would likewise augment the effects on feeding behavior and body weight as compared to unilateral injections, real-time food intake was assessed in a home-cage environment employing the Promethion system (SableSystem, Reno, NV). Food and water intake as well as body weight were assessed for up to 6 hr after CNO injection, the time period where most of the responses to this activation occur (*Whissell et al., 2016*).

Cumulative food intake was significantly increased in response to bilateral LHA→hM3Dq CART neuron activation in the CART-deficient LHA→hM3Dq *Cartpt*<sup>cre/cre</sup> mice (*Figure 10A*), and even more so in the presence of endogenous CART in LHA→hM3Dq *Cartpt*<sup>cre/+</sup> mice (*Figure 10B*). Interestingly, the increase in the amount of food consumed paralleled the increased interaction of the mice with the food hopper (*Figure 10C,D*). More importantly, although LHA→hM3Dq CART neuron activation had no effect on water intake in *Cartpt*<sup>cre/cre</sup> mice (*Figure 10E*), CNO injection in *Cartpt*<sup>cre/+</sup> mice led to a marked increase of water intake (*Figure 10F*), suggesting a critical role for LHA CART also in inducing drinking behavior. Analysis of the interaction with the water bottle revealed that, while *Cartpt*<sup>cre/cre</sup> mice showed a modest and short-lived increase in interaction in response to CNO (*Figure 10G*), CART<sup>cre/+</sup> mice demonstrated more frequent interactions following CNO compared with saline injection (*Figure 10H*). Body weight was increased in response to CNO treatment only in the presence of endogenous CART in *Cartpt*<sup>cre/+</sup> mice (*Figure 10I*), consistent with the marked increases in food and water intake, while no significant difference in body weight was

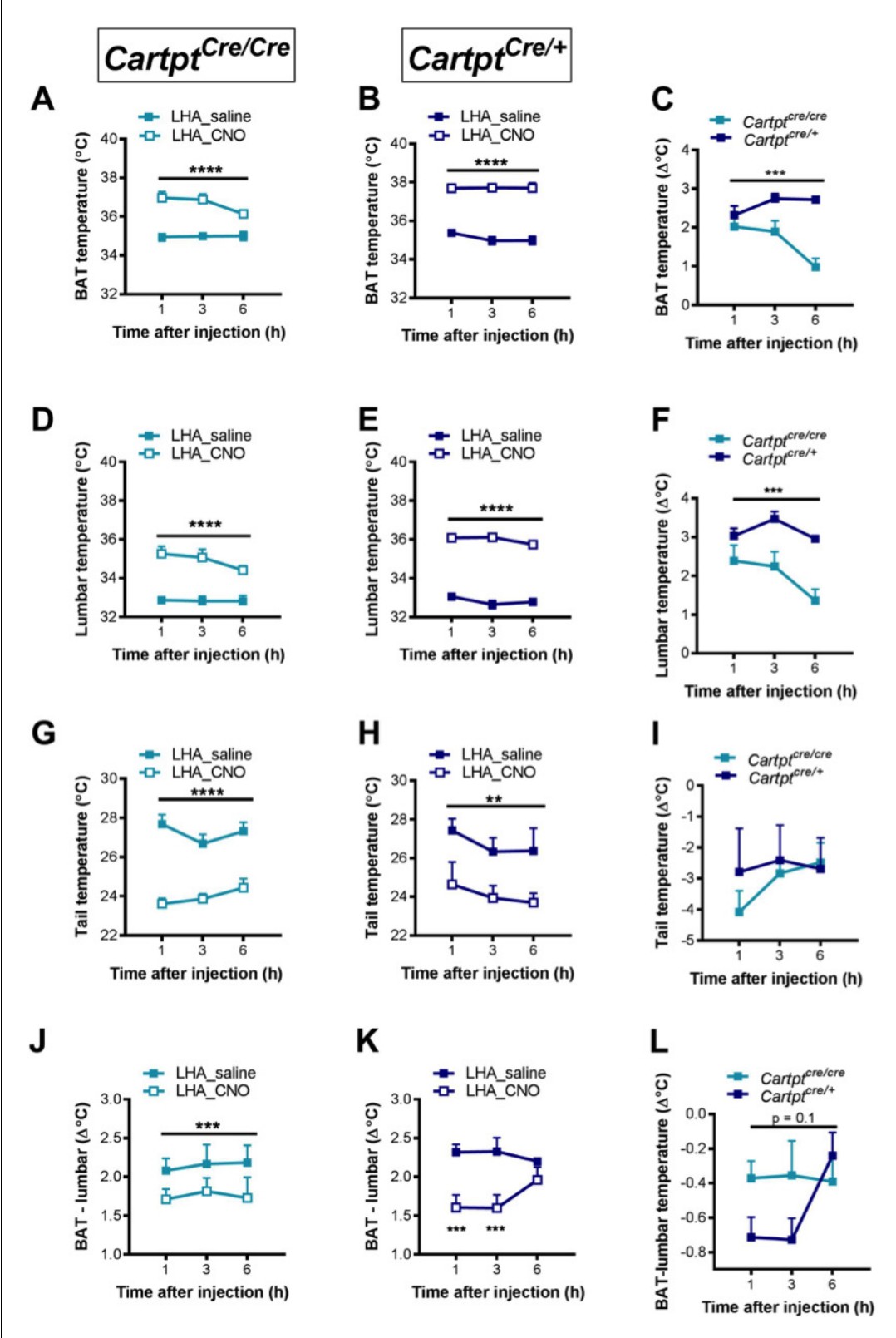

**Figure 9.** CNO-mediated bilateral activation of hM3Dq-containing LHA CART neurons leads to increased body skin temperature in *Cartpt*$^{cre/cre}$ and *Cartpt*$^{cre/+}$ mice. Temperatures of the interscapular brown adipose tissue (BAT) (**A, B**), the lumbar back (**D,E**), the tail (**G,H**) and differences between BAT and lumbar temperatures (**J, K**) measured by high-sensitivity infrared imaging 1, 3 and 6 hr after i.p. injection of saline or CNO. The difference of

*Figure 9 continued on next page*

*Figure 9 continued*

these parameters between *Cartpt^{cre/cre}* and *Cartpt^{cre/+}* mice was assessed by subtracting the value of CNO from that of saline at the corresponding time point for each mouse (C,F, I, L). Data are means ± SEM. n = 6–7; *p≤0.05, **p≤0.01, ***p≤0.001, ****p≤0.0001 for saline versus CNO treatments.

DOI: https://doi.org/10.7554/eLife.36494.011

The following figure supplement is available for figure 9:

**Figure supplement 1.** CNO does not exert any effects on body skin temperature in the absence of hM3Dq-expression in *Cartpt^{cre/cre}*, *Cartpt^{cre/+}* and wild type mice.

DOI: https://doi.org/10.7554/eLife.36494.012

observed in *Cartpt^{cre/cre}* mice (*Figure 10J*). That there are no non-specific effects of CNO administration itself is also shown in control mice (*Figure 10—figure supplement 1*).

## CART neurons of the LHA represent a heterogeneous population co-expressing various neurotransmitters affecting energy homeostasis

In order to characterize the neuronal signature of LHA CART neurons in more detail, we performed colocalisation experiments using immunohistochemistry for various genes known to be expressed in this area combined with the red fluorescence of hM3Dq expressing CART neurons. Interestingly, while no extensive colocalisation was detected some MCH and GAD67 positive neurons showed overlap with the CART neurons expressing the DREADD vector (*Figure 11A,B*), while on the other hand orexin was not found to be colocalised with CART (*Figure 11C*).

To further extend this analysis, we also employed ribosomal affinity purification technology (TRAP). For this, we crossed our *Cartpt^{cre/+}* mice onto B6;129S4-Gt(ROSA)26Sortm9(EGFP/Rpl10a) Amc/J mice in which the GFP-tagged ribosomal protein (GFP-L10a) is expressed upon *Cre-recombinase* activation, and we refer to this new line as CART-TRAP mice in which the GFP-L10a ribosomal protein is solely produced in *Cartpt*-expressing neurons (*Figure 12A*). Consistently, qPCR performed on RNA isolated by immunoprecipitation (IP) with an anti-GFP antibody against the GFP-tagged ribosomal unit from LHA preparations of these mice confirmed an enrichment of *Cartpt* transcript in the immunoprecipitated (IP) RNA relative to the corresponding unprocessed RNA (Input) sample (*Figure 12B*). Using a candidate approach we tested also for the enrichment or de-richment of known neurotransmitters and neuropeptides (*Allison et al., 2015*) that are involved in energy homeostasis control including MCH (*Pmch*), orexin (*Hcrt*), GABAergic markers (*Gad1, Gad2, Slc32a1*), neurotensin (*Nts*), the leptin receptor (*Lepr*) and galanin (*Gal*). Interestingly, we observed higher expression of most of the markers of the study in the IP population relative to the input sample (*Figure 12B*), except *Gal*, which was de-riched in the GFP-IP RNA. These results suggest that CART neurons in the LHA are highly heterogeneous and several subgroups of CART neurons exist that do co-localise with various different neuropeptides.

## Discussion

The present study demonstrates the critical and distinct roles of CART neurons in the Arc and LHA in the control of feeding behavior and energy homeostasis, and more importantly represents the first report to dissect the specific contribution of CART to these processes. Activation of Arc CART neurons leads to significant decreases in energy expenditure, locomotion and body temperature, and these effects are solely due to the contribution of CART since they are absent in Arc→hM3Dq *Cartpt^{cre/cre}* mice that are devoid of endogenous CART. Furthermore, while food intake is not affected by activation of intact Arc CART neurons, activation of these neurons in the absence of CART leads to an increase in feeding, suggesting an inhibitory role of CART in these neurons by opposing the orexigenic action of other co-expressing neurotransmitter(s). In contrast, activation of LHA CART neurons results in a marked increase in energy expenditure, locomotion, body temperature and feeding. Although these effects induced by LHA CART neuronal activation can be achieved regardless of the presence or absence of CART, the presence of CART allows these responses to occur under lower level of activation (i.e. unilateral activation) and at a greater magnitude than in the absence of CART, indicating that CART is the predominant neurotransmitter responsible for these effects.

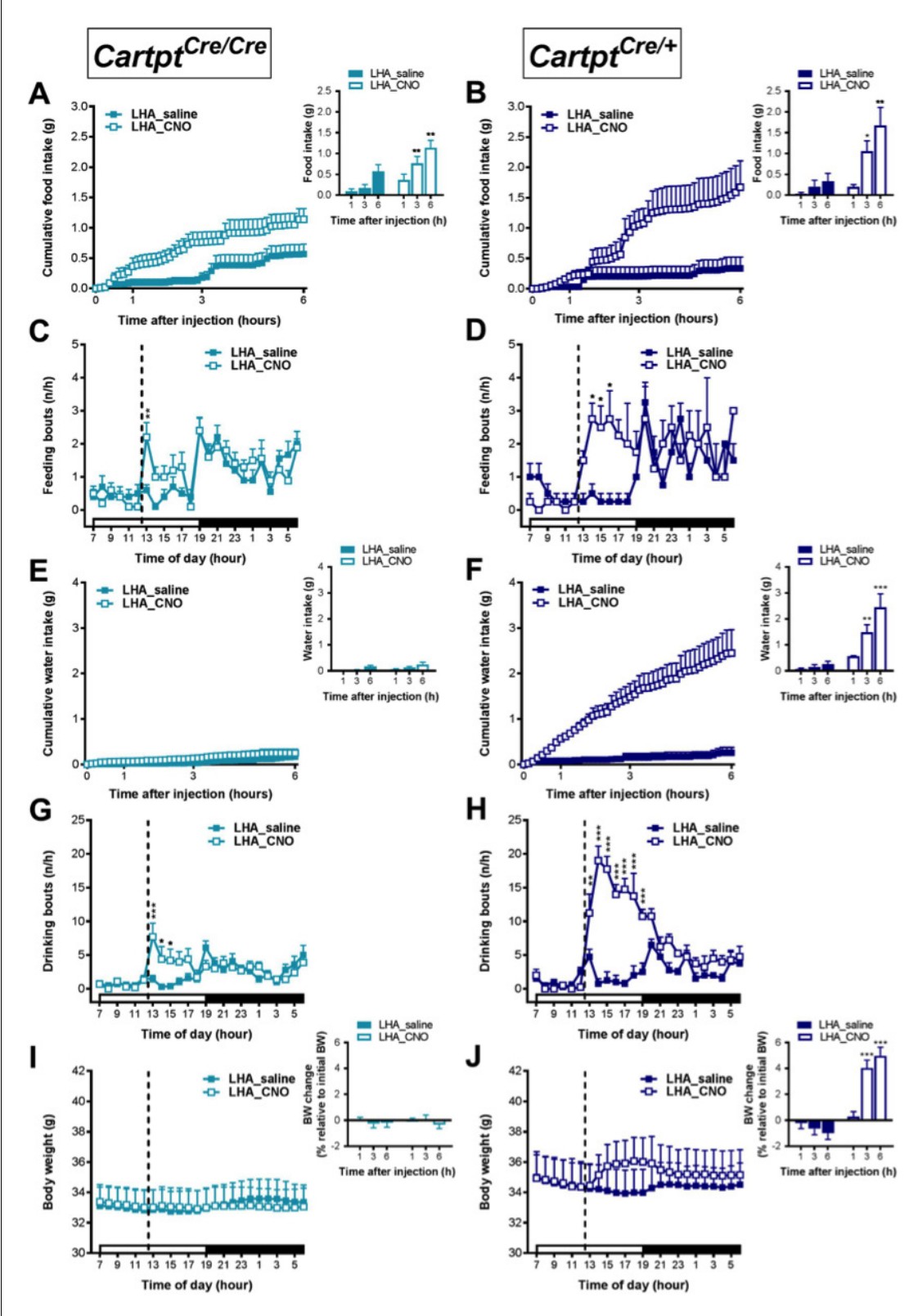

**Figure 10.** CNO-induced bilateral activation of hM3Dq-positive CART neurons in the LHA leads to an increase in ingestive behavior and body weight in heterozygous *Cartpt*<sup>cre/+</sup> mice. Cumulative food intake during fed state 6 hr following saline or CNO injection in bilateral LHA→hM3Dq *Cartpt*<sup>cre/cre</sup> and *Cartpt*<sup>cre/+</sup> mice (A, B) as well as interaction with the food hopper (C, D). Cumulative water intake during fed state 6 hr following saline or CNO injection in bilateral LHA→hM3Dq *Cartpt*<sup>cre/cre</sup> and *Cartpt*<sup>cre/+</sup> mice (E, F) and interaction with the water bottle (G, H). Body weight before and after injection of
*Figure 10 continued on next page*

*Figure 10 continued*

saline or CNO (indicated by dotted line) and corresponding body weight change at 1, 3 and 6 hr in proportion to pre-treatment body weight (**I**, **J**) measured in *Cartpt*<sup>cre/cre</sup> and *Cartpt*<sup>cr e/+</sup> LHA→hM3Dq mice receiving saline or CNO injection. Open and filled horizontal bars indicate the light and dark photoperiods, respectively. Dotted lines indicate the time of i.p. injection of saline or CNO. Data are means ± SEM. n = 6–7; *p≤0.05, **p≤0.01, ***p≤0.001 for saline versus CNO treatments.

DOI: https://doi.org/10.7554/eLife.36494.013

The following figure supplement is available for figure 10:

**Figure supplement 1.** CNO does not exert any effects on ingestive behavior and body weight in the absence of hM3Dq-expression in *Cartpt*<sup>cre/cre</sup>, *Cartpt*<sup>cre/+</sup> and wild type mice.

DOI: https://doi.org/10.7554/eLife.36494.014

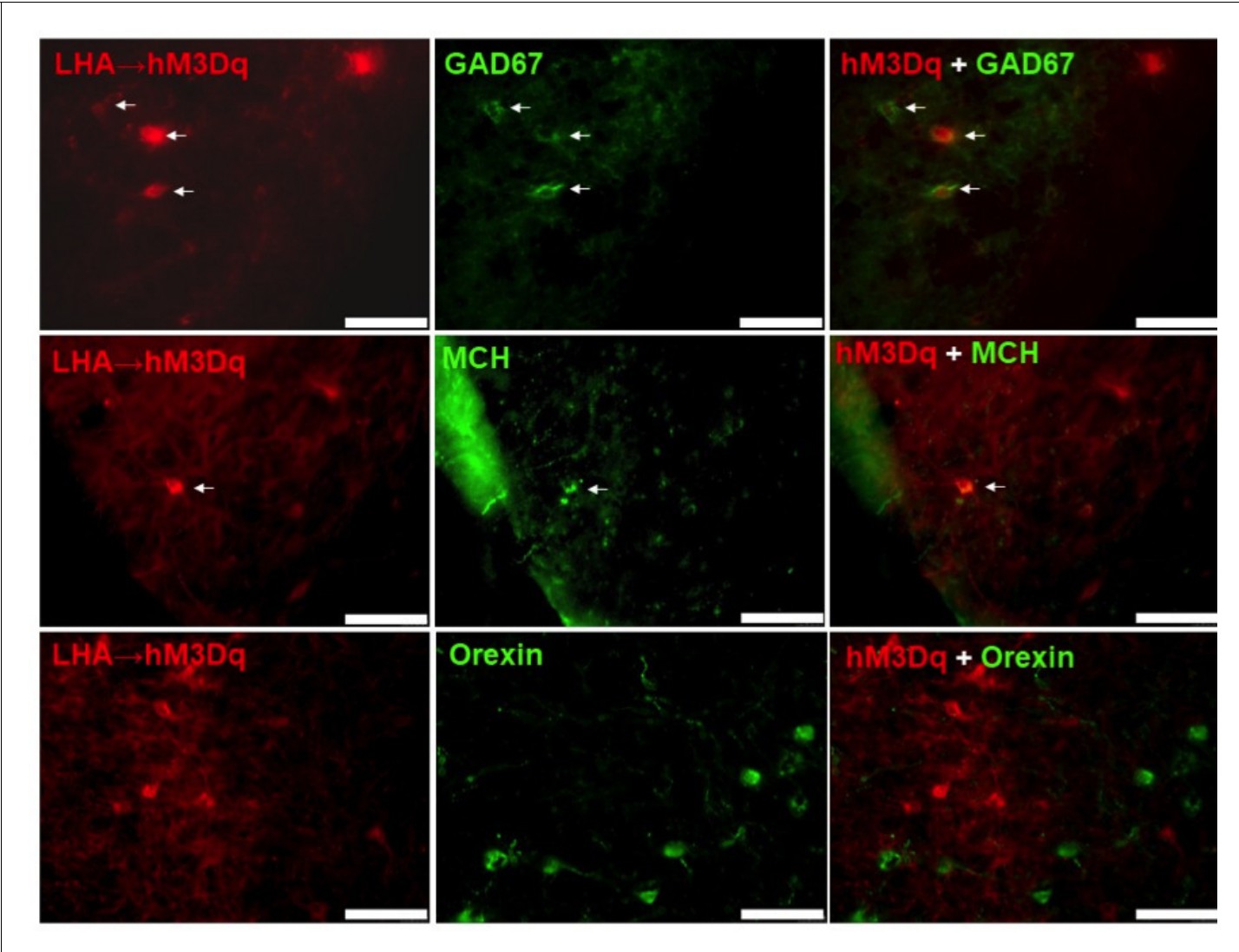

**Figure 11.** hM3Dq-mCherry-positive CART neurons co-express GAD67 and MCH, but not orexin. Fluorescence micrographs of coronal brain sections from respective LHA→hM3Dq *Cartpt*<sup>cre/+</sup> mice, showing partial co-expression of the mCherry reporter with GAD67 (**A**), and MCH (**B**) (white arrows, right panels), while no co-expression could be observed with orexin (**C**) (right panel). Scale bar = 75 µm.

DOI: https://doi.org/10.7554/eLife.36494.015

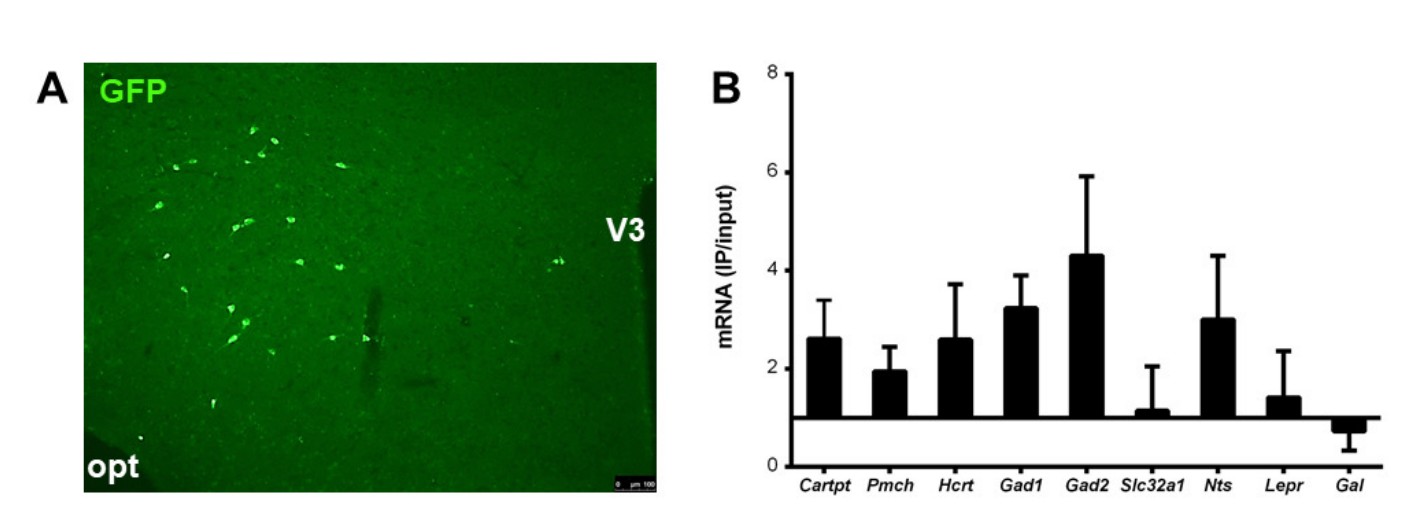

**Figure 12.** CART neurons in the LHA are highly heterogeneous and co-localise with various neurotransmitters involved in energy homeostasis. Representative fluorescence micrograph showing fluorescent CART neurons in the LHA of a CART-TRAP mouse (**A**). Enrichment (>1) or derichment (<1) of various neurotransmitter transcripts in the immunoprecipitated (IP) mRNA relative to input mRNA of the LHA of CART-TRAP mice (**B**). Data are means ± SEM. n = 3. Opt, optical tract; V3, third ventricle.
DOI: https://doi.org/10.7554/eLife.36494.016

CART is classically considered to be an anorectic and catabolic neuropeptide, given that i.c.v. administration of CART decreases food intake (*Kristensen et al., 1998*) and *Cartpt*[-/-] mice display increased body weight gain and adipose mass (*Lau et al., 2016*). However, the i.c.v delivery of CART triggers an uncoordinated and widespread simultaneous activation of CART signaling not reflecting the normal release sequence caused by physiological processes such as hunger or satiety. Similarly, global CART gene KO does not necessarily reveal the specific contributions of different populations of CART neurons in the hypothalamus and can also be plagued by developmental induced compensation.

Furthermore, typical neuromodulators of energy balance exert opposing effects on food intake and energy expenditure in order to enable homeostasis (e.g. increasing food intake, while concurrently reducing energy expenditure in response to energy deficiency) (*Bray, 2000*). Interestingly, CART in the present work does not exert these classical opposing effects on food intake and energy expenditure. Therefore, we conclude that the actions of CART in the context of this study cannot be classified as purely anabolic or catabolic. However, considering body weight alteration as the major consequence of food intake and energy expenditure, we propose that actions of CART in response to LHA CART neuron activation are largely obesigenic, while actions of CART in response to Arc CART neuron activation are mainly leptogenic. These differential effects of CART in the Arc and LHA are also mirrored by coexpression of CART with anorexigenic and orexigenic neuropeptides in the Arc and LHA, respectively (*Lau and Herzog, 2014*) as well as opposing expressional regulation of CART in these hypothalamic nuclei in response to orexigenic and anorexigenic signals (*Sergeyev et al., 2001*; *Yu et al., 2008*). Thus, high fat diet-induced obese mice have reduced *Cartpt* expression in the Arc, but increased *Cartpt* expression in the LHA compared to mice that are resistant to high fat-induced obesity (*Yu et al., 2008*). Similarly, inflammation-associated anorexia is associated with increased *Cartpt* expression in the Arc, whereas *Cartpt* expression in the LHA is decreased (*Sergeyev et al., 2001*).

In contrast to the regulation of energy balance, changes in ambient temperature typically affect food intake and energy expenditure in the same direction, thus lower ambient temperatures increase energy expenditure through adaptive thermogenesis as well as food intake in order to maintain energy homeostasis (*Lowell and Spiegelman, 2000*). The Arc has been demonstrated to be an important regulator of energy balance in response to changes in ambient temperature (*Suwanapaporn et al., 2017*) and in line with this notion we have previously reported an increase in

Arc *Cartpt* expression at thermo-neutral conditions as compared to room temperature (*Lau et al., 2016*). The observed increase in Arc *Cartpt* expression was associated with decreased energy expenditure, physical activity as well as food intake at thermo-neutral conditions (*Lau et al., 2016*). The results of the current study confirm this previous association, and provide novel functional evidence for an important role for Arc CART that might be of importance for regulating energy balance in response to varying ambient temperatures.

The results presented here are also in accordance with our previous study, where we observed reduced body weight gain following reintroduction of CART into the Arc of otherwise CART-deficient mice (*Lau et al., 2018*). While the weight loss due to reintroduction of CART in the Arc of CART-deficient mice was associated with an increase in energy expenditure with unchanged feeding behavior (*Lau et al., 2018*), the CART-dependent Arc→hM3Dq CART neuron activation in *Cartpt*-*cre/+* mice attenuated the weight gain and increased food intake otherwise seen in *Cartpt*^cre/cre^ mice, indicating differential modes of action of Arc CART in response to neuronal activation versus chronic overproduction. Interestingly, similar to the results from the Arc CART groups, differential effects on energy expenditure were induced in response to LHA→hM3Dq CART neuron activation as compared to CART reintroduction in the LHA, which led to decreased energy expenditure (*Lau et al., 2018*), again indicating the context-dependent actions of CART. As the reintroduction of CART was associated with changes in mRNA expression of hypothalamic peptides that likely contributed to the overall effects on energy homeostasis (*Lau et al., 2018*), it can be concluded that the present study reveals the actions of CART in response to acute neuronal activation without confounding expression changes in other neuropeptides, which would otherwise occur in response to chronic changes in CART expression. In addition, it has been recently demonstrated that the effects of spontaneous neurotransmitter release can remarkably differ from an evoked release (*Rau and Hentges, 2017*).

Another interesting observation of this study is that Arc→hM3Dq CART neuron activation induces an increase in food intake in the absence of endogenous CART. Considering that CART is partly co-expressed with the anorexigenic POMC in the Arc, this is surprising (*Vrang, 2006*). While pharmaco-genetic activation of POMC-expressing neurons has been reported to decrease food intake, a chronic stimulation of Arc POMC neurons was required to induce this effect (*Zhan et al., 2013*). However, recent single-cell gene expression analysis revealed a larger heterogeneity of neuronal cell types in both the Arc and the LHA than previously appreciated (*Mickelsen et al., 2017*; *Lam et al., 2017*). Specifically, the orexigenic AgRP neurons in the Arc also express, low, but detectable levels of CART (*Lam et al., 2017*), suggesting that Arc→hM3Dq CART neurons reside not solely with the anorexigenic POMC neurons, but also to some extent with the orexigenic NPY/AgRP neurons. Thus, actions from other local neuropeptides elicited upon Arc→hM3Dq CART neuron stimulation may explain in part the elevated food intake observed in *Cartpt*^cre/cre^ mice.

Our results reveal an orexigenic effect of CART in the LHA, given that LHA CART neuronal activation induced greater orexigenic responses in the presence of CART. This finding is consistent with our previous report demonstrating orexigenic effects of LHA CART-reintroduction in *Cartpt*^cre/cre^ mice (*Lau et al., 2018*), and also in line with a reported association between elevated CART expression in the LHA and increased feeding and body weight (*Yu et al., 2008*). Moreover, while the LHA is a region known to induce food intake and behavioral activation as shown by electrical stimulation of the LHA (*Stuber and Wise, 2016*), our results of orexigenic responses induced by LHA CART neuronal activation regardless of the presence of CART suggest a pivotal role of this neuronal population in mediating the metabolic and behavioral effects of the LHA (*Stuber and Wise, 2016*). Although the CART-expressing neuronal population of the LHA has been classically reported to be GABA-ergic and mainly co-expressed with MCH (*Elias et al., 2001*), novel techniques indicate a larger heterogeneity of LHA neurons, highlighting that the precise neurochemical profiles of CART neurons and their potential diverse functions remain to be characterized in more detail (*Bonnavion et al., 2016*). For instance, LHA CART neurons have been found to co-express to a small extent at mRNA but not at the protein level with orexin (*Hcrt*) (*Mickelsen et al., 2017*; *Kosse et al., 2017*; *Mavanji et al., 2015*), another neuropeptide of the LHA that promotes food intake, physical activity and energy expenditure. Interestingly, results from our immunohistochemistry and TRAP-qPCR experiments confirm these findings (*Mickelsen et al., 2017*) showing co-localisation of orexin at the mRNA level but not at the protein level. While identifying co-localisation of a variety of other neurotransmitters, we did not see any enrichment of *Gal* and only a weak enrichment of *Slc32a1*, which is in line with a recent single-cell gene expression analysis finding genes needed for GABA

synthesis (*Gad1*, *Gad2*), but not for vesicular GABA release (*Slc32a1*) in both MCH and orexin neurons (*Mickelsen et al., 2017*). The great variety of different enriched genes in the current study indicates that CART neurons in the LHA are highly heterogeneous and many different subpopulations of CART neurons may exist contributing to the spectrum of different functions seen for these CART neurons in the LHA (*Bonnavion et al., 2016*). Importantly, LHA CART has recently been demonstrated to facilitate goal-directed behavior and motivation (*Somalwar et al., 2017*), consistent with the activity-promoting effects of CART following LHA→hM3Dq CART neuron activation observed in our study. We therefore propose, that in contrast to the function of CART in the Arc, but in line with the differential effects of CART in different neuronal population, CART enforces the reward characteristics of the LHA (*Sternson and Eiselt, 2017*).

Taken together, our study demonstrates the differential metabolic effects of endogenous CART in response to pharmacogenetic activation of neurons of the Arc or LHA. The findings from this study reinforce CART's multifaceted context-dependent and region-specific functions in the modulation of feeding, thermoregulation and energy balance. Interestingly, apparent dissociation between the effects of CART on food intake and energy expenditure was unraveled in response to neuronal activation. Specifically, the presence of CART blunted the otherwise increase in food intake and body weight gain in response to Arc→hM3Dq CART neuron activation, while at the same time decreasing locomotion and energy expenditure. Opposite effects were observed in response to LHA→hM3Dq CART neuron activation, where the presence of CART promoted food intake and ingestive behavior, while increasing locomotion and energy expenditure. From a behavioral perspective, it could also be inferred that endogenous CART promotes a state of behavioral inhibition in response to Arc→hM3Dq CART neuron activation, while promoting behavioral activation in response to LHA→hM3Dq CART neuron activation.

## Materials and methods

| Reagent type (species) or resource | Designation | Source or reference | Identifiers | Additional information |
|---|---|---|---|---|
| Strain, strain background (mouse) | B6;129S4-Gt(ROSA)26 Sortm9(EGFP/Rpl10a) Amc/J Mus musculus | JAX | IMSR Cat# JAX:024750, RRID:IMSR_JAX:024750 | |
| Strain, strain background (mouse) | C57BL/6J-Tg(Zp3-cre) 82Knw/KnwJ Mus musculus | JAX | IMSR Cat# JAX:003650, RRID:IMSR_JAX:003650 | |
| Antibody | anti-c-Fos polyclonal | Santa Cruz Biotechnologies | Santa Cruz Biotechnology Cat# sc-52-G RRID:AB_2629503 | (1:1500) |
| Antibody | anti-GAD67 monoclonal | Millipore | Millipore Cat# MAB5406 RRID:AB_2278725 | (1:500) |
| Antibody | anti-MCH polyclonal | Phoenix Pharmaceuticals | Phoenix Pharmaceuticals Cat# H-070–47 RRID:AB_10013632 | (1:1000) |
| Antibody | anti-orexin polyclonal | Santa Cruz Biotechnologies | Santa Cruz Biotechnology Cat# sc-8070 RRID:AB_653610 | (1:1000) |
| Antibody | anti-GFP | Invitrogen | Thermo Fisher Scientific Cat# A-11122 RRID:AB_221569 | (10ug/TRAP experiment) |
| Antibody | Alexa 488 secondary anti-rabbit | Molecular Probes | Molecular Probes Cat# A-11094 RRID:AB_221544 | (1:500) |
| Antibody | Alexa 488 secondary anti-goat | Molecular Probes | Molecular Probes Cat# A-11073 RRID:AB_142018 | (1:500) |

*Continued on next page*

*Continued*

| Reagent type (species) or resource | Designation | Source or reference | Identifiers | Additional information |
|---|---|---|---|---|
| Antibody | Alexa 488 secondary anti-mouse | Molecular Probes | Molecular Probes Cat# A-21202 RRID:AB_141607 | (1:500) |
| Other | Fluoroshield with DAPI | Sigma-Aldrich | Sigma-Aldrich Cat# F6057 | |
| Other | SYBR Green I Nucleic Acid Gel Stain - 10,000X concentrate in DMSO | Thermofisher | Thermofisher Cat# S7567 | |
| Other | Platinum Taq DNA Polymerase | Thermofisher | Thermofisher Cat# 10966026 | |
| Commercial assay or kit | Dynabeads Protein G for Immunoprecipitation | Invitrogen | Invitrogen Cat# 10004D | |
| Commercial assay or kit | RNeasy Micro Kit | Qiagen | Qiagen Cat# 74004 | |
| Commercial assay or kit | SuperScript III First-Strand Synthesis System | Thermofisher | Thermofisher Cat# 18080051 | |
| Chemical compound, drug | Cycloheximide | Sigma-Aldrich | Sigma-Aldrich Cat# C7698 | |
| Chemical compound, drug | Protease inhibitor | Roche | Roche Cat# 05892791001 | |
| Chemical compound, drug | RNasin Ribonuclease Inhibitors | Promega | Promega Cat# N2111 | |
| Chemical compound, drug | 1,2-diheptanoyl-*sn*-glycero-3-phosphocholine, powder (DHPC) | Sigma-Aldrich | Sigma-Aldrich Cat# 850306P | |
| Chemical compound, drug | (Z)—4-Hydroxytamoxifen | Sigma-Aldrich | Sigma-Aldrich Cat# H7904 | |
| Chemical compound, drug | Clozapine N-oxide | Sigma-Aldrich | Sigma-Aldrich Cat# C0832 | |
| Software, algorithm | GraphPad Prism 6 for Mac OS X | GraphPad Software | Graphpad Prism, RRID:SCR_002798 | |
| Software, algorithm | SPSS for Mac OS X version 16.0.1 | SPSS Inc | SPSS RRID:SCR_002865 | |
| Software, algorithm | LightCycler Software | LightCycler Software | LightCycler Software RRID:SCR_012155 | |

## Animals

All experimental and animal care procedures were approved by the Garvan Institute/St. Vincent's Hospital Animal Ethics Committee and were conducted in agreement with the Australian Code of Practice for the Care and Use of Animals for Scientific Purposes. Male mice were used for all experiments and were housed under conditions of controlled temperature (22°C) and illumination (12:12 hr light-dark cycle, lights on at 07:00 am). Mice were provided with ad libitum access to water and a standard chow diet (8% calories from fat, 21% calories from protein, 71% calories from carbohydrate, 2.6 kcal/g; Gordon's Speciality Stock Feeds, Yanderra, NSW, Australia).

## Generation of *Cartpt*[cre/cre], *Cartpt*[cre/+] and CART-TRAP mice

To investigate the effects of activation of CART expressing neurons, CART neuron-specific introduction of genetic elements in an adult-onset inducible manner was enabled through the generation of

a conditional *Cartpt-cre* knock-in mouse line on a C57/Bl6 background as described previously (*Lau et al., 2016*). In short, the targeting vector was designed to replace the *Cartpt* coding exons with the mouse *Cartpt* cDNA fused to the original *Cartpt* gene at the initiation codon, which is followed by the insertion of a flippase (Flp) recombinase-recombination target (FRT)-flanked neomycin selection cassette. A cassette carrying the tamoxifen-inducible *Cre-recombinase* gene followed by an internal ribosome entry site (IRES) that drives the expression of the mCherry red fluorescence reporter gene was inserted. *LoxP* sites were placed immediately upstream of the initiation codon of the *Cartpt* and *Cre* genes respectively, which allows the replacement of the *Cartpt* gene by *Cre*. Positively targeted ES cell clones were injected into blastocysts. Chimeric offspring were crossed to create heterozygous conditional CART mice with the floxed gene (*Cartpt^{lox/+}*), which were then bred with a germline Flp-recombinase mouse line to remove the neomycin cassette and further crossed to homozygosity to generate the conditional CART floxed knockout line (*Cartpt^{lox/lox}*). The *Cartpt^{lox/lox}* line was then crossed with C57BL/6J-Tg(Zp3-cre)82Knw/KnwJ mice that expressed *Cre* specifically in the oocytes, resulting in a rearrangement that combines the inducible *Cre* gene with the endogenous CART promoter. Subsequent crossing to homozygosity ultimately generated the novel inducible *Cartpt-cre* knock-in mouse line (*Cartpt^{cre/cre}*). Since the endogenous *Cartpt* gene has been replaced by the *Cre-recombinase* gene, the resultant *Cartpt^{cre/cre}* mice also represent a classical germline CART knockout mouse line (*Cartpt^{-/-}*), while the heterozygous *Cartpt^{cre/+}* mice do express endogenous CART. To activate the CART promoter controlled *Cre-recombinase* gene 1 µl of the active metabolite of tamoxifen, 4-hydroxytamoxifen (4-OHT) (20 mg/mL) (Sigma-Aldrich Pty Ltd, Sydney, NSW, Australia) dissolved in absolute ethanol was injected into the left lateral cerebral ventricle (LV) of 10–12 week-old *Cartpt^{cre/cre}* and *Cartpt^{cre/+}* mice. The injection coordinates relative to Bregma were –0.34 mm anteroposterior,+1.0 mm mediolateral, and –2.5 mm dorsoventral corresponding to the left LV.

For CART-TRAP mice, to activate the *Cre-recombinase* 1 µl of the active metabolite of tamoxifen, 4-OHT (20 mg/mL) (Sigma-Aldrich Pty Ltd, Sydney, NSW, Australia) was injected into the LV two weeks before the isolation of the LHA for TRAP experiments.

## Generation and expression of *Cre*-inducible AAV-hM3Dq-mCherry vector

Chemogenetic activation of CART neurons was performed by injection of a rAAV6-hSyn-DIO-hM3Dq-mCherry vector ($1 \times 10^{13}$ viral particles/mL, Addgene #44361 Bryan Roth) in an inducible manner. Upon exposure to tamoxifen, *Cre-recombinase* in *Cartpt^{crecre}* and *Cartpt^{cre/+}* mice is translocated to the nucleus of CART neurons, thereby catalyzing the inversion of the *loxP*-flanked cassette resulting in hM3Dq receptor expression in CART neurons only. Accordingly, activation of the hM3Dq-expressing CART neurons was triggered remotely by i.p. injection of CNO (0.3 mg/kg).

The same procedures were repeated in wild type mice, to control for unspecific hM3Dq receptor expression in the absence of CART neuronal *Cre-recombinase* expression. Furthermore, *Cartpt^{cre/cre}* and *Cartpt^{cre/+}* mice without injection of a hM3Dq-expressing vector were used as another control to investigate potential effects of CNO.

## Stereotaxic delivery of AAV-hM3Dq-mCherry to CART neurons in *Cartpt-cre* knock-in mice

Adult-onset central CART neuron-specific expression of the hM3Dq receptor in the Arc or LHA was achieved by intra-hypothalamic injection of the hM3Dq-mCherry viral vector in parallel with i.c.v. injection of 1 µl of the active metabolite of tamoxifen, 4-hydroxytamoxifen (4-OHT) (20 mg/mL) (Sigma-Aldrich Pty Ltd, Sydney, NSW, Australia) in *Cartpt^{cre/cre}* and *Cartpt^{cre/+}* mice. The stereotactic delivery was performed on 12-week-old *Cartpt^{cre/cre}* and *Cartpt^{cre/+}* mice as described previously (*Lau et al., 2018*). 0.75 µl of the rAAV6-hSyn-DIO-hM3Dq-mCherry vector ($1 \times 10^{13}$ viral particles/ mL, Addgene #44361 Bryan Roth) was injected at the right Arc (coordinates relative to Bregma: – 2.18 mm anteroposterior, –0.25 mm mediolateral, and –5.5 mm dorsoventral). LHA hM3Dq receptor expression was achieved by either bilateral or unilateral injection at the LHA (coordinates relative to Bregma: –1.94 mm anteroposterior, –1.0 mm mediolateral, and –5.25 mm dorsoventral). To activate the *Cre-recombinase* for mediating the transgene inversion and expression, 1 µl of the active metabolite of tamoxifen, 4-hydroxytamoxifen (4-OHT) (20 mg/mL) (Sigma-Aldrich Pty Ltd, Sydney, NSW,

Australia) dissolved in absolute ethanol was then injected unilaterally into the left lateral cerebral ventricle (LV). The injection coordinates relative to Bregma were –0.34 mm anteroposterior,+1.0 mm mediolateral, and –2.5 mm dorsoventral corresponding to the left LV. Mice were allowed two weeks of recovery with regular body weight assessment prior to any phenotypic characterization procedures.

For simplicity, the post-surgery subjects are referred to as Arc→hM3Dq *Cartpt*$^{cre/cre}$ or *Cartpt*$^{cre/+}$ and LHA→hM3Dq *Cartpt*$^{cre/cre}$ or *Cartpt*$^{cre/+}$ mice. At least 6 mice were included for each of the groups resulting in a number of 6–12 mice per site of injection and genotype.

## Characterization of food intake and body weight

To investigate the physiological effects of activating Arc→hM3Dq or LHA→hM3Dq on feeding regulation, food intake in *Cartpt*$^{cre/cre}$ and *Cartpt*$^{cre/+}$ mice (n ≥ 6 for each of 6 groups) in response to treatment of either CNO or saline control was measured. After a 2–3 weeks of recovery period after AAV-hSyn-DIO-hM3Dq-mCherry delivery, all mice were at 15 wks of age and fed a standard laboratory chow. All mice undergoing unilateral hM3Dq delivery were assessed for basal feeding behavior for 24 hr as described previously (*Lau et al., 2018*).

For measurement of 24 hr feeding, ad libitum fed mice subjected to unilateral hM3Dq delivery were i.p. injected at 5:00 pm, shortly before dark onset, with a single dose of either saline or CNO (0.3 mg/kg of BW). Food intake was measured 24 hr later at 5:00 pm, during which the same experiment was repeated immediately in a crossover design, such that mice receiving saline treatment before were i.p. injected with CNO, and vice versa. The crossover design allows each mouse to serve as both the experimental and the control subject, such internal biological control method minimizes the sample size required for achieving statistical strength.

For the bilateral injected LHA→hM3Dq cohorts, food and water intake as well as body weight were assessed within Promethion metabolic cages (Promethion$^{TM}$, Sable Systems International, NV USA). This calorimetry system consists of 8 metabolic cages (identical to home cages with bedding) each equipped with water bottles and food hoppers connected to load cells for food and water intake monitoring, and all animals had ad libitum access to standard rodent chow and water throughout the study (*Kaiyala et al., 2012*). After a 3-day run-in period in which mice were acclimatized to the chambers, mice were alternately injected with saline or CNO (0.3 mg/kg of BW) as described before. The time point of injection was, however, preponed to 12:00 pm, in order to unveil potential LHA→hM3Dq-induced increases of ingestive behavior, which could be otherwise overlaid by the physiologic increase of food intake at dark onset.

## Indirect calorimetry of energy expenditure and assessment of physical activity

To determine the metabolic effects of activating specific hypothalamic CART neuron populations, all *Cartpt*$^{cre/cre}$ and *Cartpt*$^{cre/+}$ mice with Arc→hM3Dq or LHA→hM3Dq (n ≥ 6 for each of 4 groups) were evaluated for metabolic parameters and physical activity at 17–18 wks of age. For energy metabolism, metabolic rate of the unilateral injected cohorts was measured by indirect calorimetry using an 8-chamber open-circuit calorimeter (Oxymax series; Columbus Instruments) (*Lau et al., 2018*). At 5:00 pm before dark onset, mice were i.p. injected with either saline or CNO (0.3 mg/kg of BW). Monitoring and data recording was then performed for 24 hr in the metabolic chambers, where the individually housed mice were provided ad libitum access to water and chow food in pellet form. At the end of the 24 hr recording session at 5:00 pm the next day, the same experiment was immediately repeated in a crossover design, such that mice receiving saline treatment before were i.p. injected with CNO, and vice versa. Mice were immediately placed back into the metabolic chambers after injection, and data were recorded for the next 24 hr until 5:00 pm the following day. For both 24 hr intervals, body weight was noted and food given in excess was weighed before and after the recording period for measuring daily food intake after subtraction of spillage.

For the bilateral injected cohort indirect calorimetry was performed within the Promethion metabolic cages (Promethion$^{TM}$, Sable Systems International, NV USA) (*Kaiyala et al., 2012*). As described for the assessment of food intake, alternated injections of saline or CNO (0.3 mg/kg of BW) were performed at 12:00 pm in this cohort.

## Measurement of whole body and brown adipose tissue temperatures by infrared imaging

Whole body skin temperature and skin temperature at the interscapular brown adipose tissue (BAT) were measured for both the *Cartpt*<sup>cre/cre</sup> and *Cartpt*<sup>cre/+</sup> cohorts by non-invasive high-sensitivity infrared imaging with a high-sensitivity infrared camera (ThermoCAM T640, FLIR, Danderyd, Sweden, sensitivity = 0.04°C) as described previously (*Lau et al., 2016*). Temperature measurement was performed daily for three consecutive days in 19–20-week-old mice. On the first day, all mice were injected with saline, as a habituation to the injection procedure. On day 2 and three mice were alternately i.p. injected with saline or CNO (0.3 mg/kg of BW). Temperature was measured 2 hr after injection for the unilateral injected hM3Dq cohorts. In order to get insights into the time course of the changes in skin temperature, temperature was assessed 1, 3 and 6 hr after saline or CNO in the bilateral injected LHA→hM3Dq cohort.

## Verification of the expression site and neuronal activation of hM3Dq positive cells by fluorescent microscopy and immunohistochemistry

To validate the injection sites for the AAV-hSyn-DIO-hM3Dq-mCherry and neuronal activation conferred by CNO in hM3Dq-expressing CART neurons, protein expression of mCherry and c-Fos was examined in brains collected from *Cartpt-cre* knock-in mice as described (*Shi et al., 2013*). At the completion of study, mice received an i.p. injection of either saline or CNO (0.3 mg/kg of BW) 60 min prior to sacrifice via transcardial perfusion. Each of the *Cartpt*<sup>cre/cre</sup> and *Cartpt*<sup>cre/+</sup> cohorts with Arc→hM3Dq or LHA→hM3Dq was evenly divided into two groups for the treatment of saline or CNO.

For identification of the injection sites marked by mCherry expression produced by the mCherry reporter coupled to the AAV-hSyn-DIO-hM3Dq-mCherry targeting construct, direct fluorescence microscopy was performed. For the detection of neuronal activation in hM3Dq-containing cells triggered by CNO, c-Fos expression marking neuronal activity was assessed by immunohistochemistry (*Shi et al., 2013*).

Sections were incubated overnight at room temperature with rabbit anti-c-Fos polyclonal primary antibody (Santa Cruz Biotechnology Inc, Santa Cruz, CA, USA) diluted 1:1500 in PBST with 2% normal goat serum and 1% bovine serum albumin. Following three 10 min washes in PBST, sections were incubated with anti-rabbit Alexa Fluor 488 (1:500; Molecular Probes, Eugene, OR) for 2 hr (*Shi et al., 2013*).

For visualization of Fos immunoreactivity at the sites of AAV-hM3Dq delivery, hypothalamic brain sections containing the targeted Arc or LHA regions from respective Arc→hM3Dq and LHA→hM3Dq mice were investigated using a Zeiss Axiophot microscope equipped with a ProgRes 3008 digital camera (Carl Zeiss).

In order to investigate co-expressed transmitters within the hM3Dq-expressing CART neurons immunohistochemistry was also performed for GAD67 (mouse anti-GAD67 monoclonal primary antibody, MAB5406, 1:500, Chemicon International, Temecula, CA, USA), secondary anti-mouse Alexa Fluor 488 antibody (1:500, Molecular Probes, Eugene, OR), MCH (rabbit-anti MCH, 1:1000, Phoenix Pharmaceuticals, Inc., Belmont, CA, USA), secondary anti-rabbit Alexa Fluor 488 antibody (1:500, Molecular Probes, Eugene, OR) and orexin (goat anti-orexin polyclonal primary antibody, sc-8070, 1:1000, Santa Cruz Biotechnology Inc, Santa Cruz, CA, USA), secondary anti-goat Alexa Fluor 488 antibody (1:500, Molecular Probes, Eugene, OR).

## Ribosome affinity purification technology (TRAP)

TRAP experiment was performed based on the previously published protocol with some modifications (*Heiman et al., 2014*; *Loh et al., 2017*; *Zhou et al., 2013*). Reagents used here were the same as those reported previously (*Loh et al., 2017*), unless stated otherwise. Freshly excised whole brain tissue was placed into chilled dissection buffer supplemented with fresh 100 ug/mL of cycloheximide (CHX; Sigma). Subsequently the LHA was micro-dissected and immediately processed further for the TRAP experiment. The dissected brain tissues were homogenized with a hand pestle mixer (Argos Technologies) in 500 µL of tissue lysis buffer supplemented with fresh protease inhibitor (Roche), CHX and RNAsin (Promega), and then incubated at 4°C for 5 min. Homogenates were centrifuged for 10 min at 2000 x g at 4°C to remove pellet nuclei and cell debris, and then 50 µL of the 10% NP-

40 (10 %vol/vol; Biochemica) and DHPC (300 mM; Avanti Polar Lipids) were added to the supernatant and mixed gently by inverting the mixture for 10 times. After incubation on ice for 5 min, the lysate was centrifuged for 10 min at 13,000 x g. 20% of the lysate was kept as input. For the preparation of the antibody-beads, 50 µL protein G Dynal magnetic beads (Invitrogen) were washed three times with 0.15 M KCl buffer at room temperature (RT), and then 5 uL of anti-GFP antibody (2 µg/µL; Invitrogen) was added into the beads and incubated with beads suspended in 275 uL of KCl buffer for 1 hr at RT with mild end-to-end rotation. Next, the antibody-bound beads were collected by using the magnetic rack and washed three times with 0.15 M KCl buffer before use. The beads were then mixed with the cell-lysate supernatant, and the mixture was incubated at 4°C with mild end-to-end rotation overnight. Beads were subsequently collected on a magnetic rack, washed three times with 0.35 M KCl buffer supplemented with fresh CHX at 4°C, and immediately placed in 350 µL of RNA lysis buffer (RLT; Qiagen) supplemented with 10% of 2-mercaptoethanol at RT and incubated for 5 min. Next, the RLT-containing RNA was purified with the RNeasy microKit (Qiagen) following manufacturer's protocol. DNase digestion step was included in the purification. RNA quantification and purity were confirmed by NanoDrop Spectrophotometers. 40 ng of RNA was reverse transcribed into cDNA using the SuperScript III First-Strand Synthesis System (Thermo Fisher Scientific). RT-qPCR using primers for *Cartpt, Gad1, Gad2, Gal, Hcrt, Lepr, Nts, Pmch, Slc32a1* as indicated in *Supplementary file 1* was carried out in samples prior (input) and after the immunoprecipitaion (IP) using the LightCycler (Light-Cycler 480 Real-Time PCR system, Roche Applied Science, Germany), SYBR Green I (Molecular Probes) and Platinum Taq DNA Polymerase (Invitrogen). The previously described PCR condition was used in all the RT-qPCR experiments, 94°C for 30 s, 62°C for 30 s, 72°C for 20 s for 40 cycles (*Ip et al., 2014*). Expression of the gene was normalised to the expression of housekeeping gene *Actb* and expressed as relative to input values. Sequences of the oligonucleotide primers used for these PCRs can be found in *supplementary file 1*.

## Statistical analysis

All data are presented as means ± SEM. Differences amongst mouse groups of various genotypes and treatments were assessed by ANOVA or repeated-measures ANOVA combined with Bonferroni post-hoc analysis where appropriate. Energy expenditure, RER and physical activity over the continuous 24 hr period were averaged for the whole 24 hr period, as well as the 12 hr light and dark phases individually. Statistical analyses were performed with GraphPad Prism 6 for Mac OS X (GraphPad Software, Inc. CA, USA) and SPSS for Mac OS X version 16.0.1 (SPSS Inc., Chicago, IL, USA). Statistical significance was defined as p value$\leq$0.05.

## Acknowledgements

This research was supported by the National Health and Medical Research Council of Australia (NHMRC #1081290), a Research Fellowship to H.H. (NHMRC #118775), a Postgraduate Scholarship to J.L. (NHMRC #1039847), and the Austrian Science Fund (FWF grant J-3814) to A.F.

# Additional information

### Funding

The funders had no role in study design, data collection and interpretation, or the decision to submit the work for publication.

### Author contributions

Aitak Farzi, Conceptualization, Data curation, Formal analysis, Investigation, Writing—original draft; Jackie Lau, Conceptualization, Data curation, Formal analysis, Investigation, Visualization, Writing—original draft; Chi Kin Ip, Lei Zhang, Data curation, Formal analysis, Methodology, Writing—review and editing; Yue Qi, Data curation, Methodology, Writing—review and editing; Yan-Chuan Shi, Formal analysis, Methodology, Writing—review and editing; Ramon Tasan, Günther Sperk, Investigation, Methodology, Writing—review and editing; Herbert Herzog, Conceptualization, Resources, Formal analysis, Supervision, Funding acquisition, Methodology, Writing—original draft, Project administration

## Author ORCIDs

Aitak Farzi  https://orcid.org/0000-0001-9606-3871
Jackie Lau  https://orcid.org/0000-0003-2164-1029
Yan-Chuan Shi  https://orcid.org/0000-0002-8368-6735
Günther Sperk  https://orcid.org/0000-0001-6561-8360
Herbert Herzog  http://orcid.org/0000-0002-1713-1029

## Ethics

Animal experimentation: All experimental and animal care procedures were approved by the Garvan Institute/St. Vincent's Hospital Animal Ethics Committee (permit AEC #17/01) and were conducted in agreement with the Australian Code of Practice for the Care and Use of Animals for Scientific Purposes.

## Decision letter and Author response

Decision letter https://doi.org/10.7554/eLife.36494.020
Author response https://doi.org/10.7554/eLife.36494.021

## Additional files

### Supplementary files

• Supplementary file 1. Sequences of oligonucleotide primers used in qPCR
DOI: https://doi.org/10.7554/eLife.36494.018
• Transparent reporting form
DOI: https://doi.org/10.7554/eLife.36494.019

### Data availability

All original data generated or analysed are included in the manuscript

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
