## [Decision Letter]

Thank you for submitting your article "Opposing effects of arcuate nucleus and lateral hypothalamic CART neurons on energy expenditure" for consideration by *eLife*. Your article has been reviewed by three peer reviewers, and the evaluation has been overseen by a Reviewing Editor and a Senior Editor. The following individual involved in review of your submission has agreed to reveal his identity: Michael A Cowley (Reviewer #3).

The reviewers have discussed the reviews with one another and the Reviewing Editor has drafted this decision to help you prepare a revised submission.

Summary:

Farzi and colleagues took advantage of the chemogenetic DREADD technology to generate original results, which further delineate the function of hypothalamic CART and CART neurons in food intake and energy expenditure. They report opposite effects following depolarization of Arc CART versus LHA CART, with Arc CART neurons decreasing, while LHA CART increasing, energy expenditure and physical activity, effects that are attenuated in the absence of CART. They also show different effects on food intake following activation of those CART populations. They conclude that activation of Arc CART neurons produces anorexigenic but catabolic effects, while activation of LHA CART produces orexigenic but catabolic effects. Overall, the manuscript reads well and presents novel results clarifying CART neurons' role in energy metabolism. However, the reviewer enthusiasm is limited by the fact that the study is still descriptive in nature and is at some extent confusing to follow.

Essential revisions:

Figure 1 is not conveying the point they are trying to make. The authors could add additional rostro-caudal pictures allowing a better reflection of the infection pattern. Moreover, the pictures show a very lateral injection pattern, especially for the Arc. The c-Fos photomicrographs are not very convincing and could be improved.

Injecting mice with empty vectors is not the only control needed. An additional group of WT receiving DREADD viruses and CNO would be welcome.

Activation of Arc CART neurons (in *Cartpt^cre/+^*mice – I assume this means that they had some CART?) reduced energy expenditure, skin temperature over intrascapular BAT depots, and physical activity. The BAT temperature data suggest that there may have been changes in sympathetic nervous system outflow. In contrast, the effect of neuronal activation on food intake and body weight was not seen in the *Cartpt^cre/+^*mice, but was seen in the *Cartpt^cre/cre^*mice – the effect was a reduction in food intake.

The authors need to comment on what a skin temperature change in the intrascapular regions really means. How could BAT temperature decrease? BAT is a thermogenic organ, and as such would not be expected to lower than skin temperature. A better characterization of BAT thermogeneic capacity/activity would be helpful. In addition, did the authors assess the levels of thermogenic markers such as UCP1?

The mouse models description is very confusing: what germline CART KO was used? What dose of tamoxifen and when was it injected? What oocyte expressing-Cre line did authors use? Was it a ZP3-cre? Tamoxifen-inducible? Information about the source and background should be included. Was icv tam or hydroxyl tam used? The gene nomenclature does not follow the guidelines of the journal, by example, CART-cre should be *Cartpt^cre/cre^.* This issue makes it very hard to interpret the data. This should be made clearer in a revision.

Apparently, the energy balance measurements were only acutely performed, which limits the speculation on the effects of CART neurons in the regulation of energy balance. A sub-chronic (14 days) treatment (with CNO in the drinking water) in, for instance the LHA model, would have possibly been a nice addition. The LHA model that allowed for bilateral depolarizations was apparently the most interesting one and could have been further exploited.

Neuroanatomical experiments to further describe the identity of CART neurons (particularly in the LHA) would have added useful information to determine whether the LHA CART neurons are part of circuitries genuinely involved in energy homeostasis regulation.

Activation of CART neurons in the LHA of *Cartpt^cre/+^*mice increased energy expenditure and physical activity, but did not alter dermal temperature over the scapulae. It also increased food intake. The effect appeared to be stronger in mice with bilateral injection of the activator construct, which is not surprising.

---

## [Author Response]

Essential revisions:Figure 1 is not conveying the point they are trying to make. The authors could add additional rostro-caudal pictures allowing a better reflection of the infection pattern. Moreover, the pictures show a very lateral injection pattern, especially for the Arc. The c-Fos photomicrographs are not very convincing and could be improved.

We have expanded this analysis and provide now additional picture that more clearly demonstrate the points we like to convey. These figures include rostro-caudal pictures reflecting the infection pattern of the Arc and LHA injection sites (Figure 1A,B) as well as improved c-Fos micrographs showing CNO-induced c-Fos expression in hM3Dq-mCherry-positive CART neurons. The expression pattern are in agreement with our previous in situ analysis (https://doi.org/10.1016/j.npep.2016.03.006) and also reflects our cautious approach to aim our injections on the fringe of the Arc to let the viral vector to diffuse into it rather than causing major tissue damage through the needle track. Moreover, the significant alterations seen in the different physiological readouts in the injected versus the controls are testimony for the accuracy of our viral delivery.

Injecting mice with empty vectors is not the only control needed. An additional group of WT receiving DREADD viruses and CNO would be welcome.

We fully agree and actually these controls were already included in our original manuscript (Supplementary Figures 1-3). These negative controls are now described in more detail in the Results section and the lack of responses in WT mice receiving DREADD viruses and CNO are presented in Figure 2—figure supplement 1, Figure 9—figure supplement 1 and Figure 10—figure supplement 1.

Activation of Arc CART neurons (in Cartpt^cre/+^ mice – I assume this means that they had some CART?) reduced energy expenditure, skin temperature over intrascapular BAT depots, and physical activity. The BAT temperature data suggest that there may have been changes in sympathetic nervous system outflow. In contrast the effect of neuronal activation on food intake and body weight was not seen in the Cartpt^cre/+^ mice, but was seen in the Cartpt^cre/cre^ mice – the effect was a reduction in food intake.The authors need to comment on what a skin temperature change in the intrascapular regions really means. How could BAT temperature decrease? BAT is a thermogenic organ, and as such would not be expected to lower than skin temperature. A better characterization of BAT thermogeneic capacity/activity would be helpful. In addition, did the authors assess the levels of thermogenic markers such as UCP1?

We appreciate the reviewers comment. Indeed, one has to be cautious when interpreting measurements of surface (skin) temperature covering the inter-scapular brown adipose tissue (BAT) as a measure of BAT thermogenesis. In fact, temperature differences between the inter-scapular and lumbar areas from the same animal (TBAT-TBack) are a better measure of the thermogenic contribution of inter-scapular brown adipose tissue. For that reason, we now also present this TBAT-TBack temperature difference in response to Arc CART neuron activation (Figure 3). As there is no significant difference between the changes in surface temperature of the BAT and lumbar regions, we conclude that the observed decrease in surface (skin) temperature is not due to a specific change in BAT thermogenic activity. Furthermore, the observation that the decrease in temperature is paralleled by a decrease in physical activity suggests that decreased activity related thermogenesis may at least in part contribute to the observed temperature changes. This aspect is now also addressed in the Results section.

The mouse models description is very confusing: what germline CART KO was used? What dose of tamoxifen and when was it injected? What oocyte expressing-Cre line did authors use? Was it a ZP3-cre? Tamoxifen-inducible? Information about the source and background should be included. Was icv tam or hydroxyl tam used? The gene nomenclature does not follow the guidelines of the journal, by example, CART-cre should be Cartpt^cre/cre^. This issue makes it very hard to interpret the data. This should be made clearer in a revision.

We have expanded the description of the different models in the Materials and methods section and include additional citations describing the original generation of the different models. In short, all mice are on a C57Bl/6 background. To generate the CART Cre knock-in line, we crossed our mice with a floxed CART gene that downstream also contained an inducible Cre-recombinase cassette with a germline oozyte (ZP3-Cre) line resulting in the removal of the CART coding sequence and its simultaneous replacement with the downstream inducible Cre-recombinase gene, which now is under the control of the endogenous CART promoter. The germline CART KO mice were generated by crossing these heterozygous CART Cre knock-in mice to homozygousity. To activate the CART promoter controlled Cre-recombinase gene the active hydroxyl tamoxifen was injected icv.

Apparently, the energy balance measurements were only acutely performed, which limits the speculation on the effects of CART neurons in the regulation of energy balance. A sub-chronic (14 days) treatment (with CNO in the drinking water) in, for instance the LHA model, would have possibly been a nice addition. The LHA model that allowed for bilateral depolarizations was apparently the most interesting one and could have been further exploited.

Our study was designed according to standard procedures and technologies used and published previously in *eLife* (Burke et al., 2017, 10.7554/eLife.22848) and in other journals (Coutinho et al., 2017, 10.3390/ijms17071065). While the potential chronic treatment with CNO is a possible addition we believe this would not add significant additional information to the already extensive evaluation of these models.

Neuroanatomical experiments to further describe the identity of CART neurons (particularly in the LHA) would have added useful information to determine whether the LHA CART neurons are part of circuitries genuinely involved in energy homeostasis regulation.

To further characterise the nature of the LHA CART neurons we employed IHC colocalisation experiments as well as ribosomal affinity purification technology (TRAP). For this, GFP-L10a fusion protein encoding Rosa26CAG-EGFP (TRAP) mice were crossed onto our *Cartpt^cre/+^*mice to generate the *CART^cre/+^;TRAP^lox/lox^* mice (CART-TRAP), where the ribosomal L10a-GFP fusion protein is only produced in CART neurons (Figure 12A).

qPCR performed on mRNA isolated by immunoprecipitation with an anti-GFP antibody against the GFP-tagged ribosomal unit from LHA preparations of these mice confirmed a significant enrichment of *Cartpt* transcript in the immunoprecipitated (IP) RNA relative to the corresponding unprocessed RNA (Input) sample (Figure 12B). Using a candidate approach, we tested for the enrichment of known markers that are involved in energy homeostasis control including MCH and galanin, GABAergic markers and neurotensin.

Interestingly, there was overlap with all of these other neuropeptides except galanin, suggesting that CART neurons in the LH are highly heterogeneous and co-localise with various neuropeptides within the LHA. This great heterogeneity in CART neurons could be one of the reasons for the great spectrum of functional output from these LHA CART neurons.

Activation of CART neurons in the LHA of Cartpt^cre/+^ mice increased energy expenditure and physical activity, but did not alter dermal temperature over the scapulae. It also increased food intake. The effect appeared to be stronger in mice with bilateral injection of the activator construct, which is not surprising.

While it may not be surprising that the bilateral activation shows a stronger effect it actually is consistent with the observed heterogeneity of the CART neurons in the LH where probably only a subset of them contributes to this effect and this maybe on the threshold when they are only unilaterally activated.

Furthermore, the fact that specific activation of LHA CART neurons in the presence of endogenous CART increases EE but does not alter BAT surface temperature suggests that this may be driven by the increase in physical activity and the fuel used for this is directly sourced from the increased food intake.